# The Burden of Interactive Alignment with Inconsistent Preferences

**Ali Shirali**
UC Berkeley

## Abstract

From media platforms to chatbots, algorithms shape how people interact, learn, and discover information. Such interactions between users and an algorithm often unfold over multiple steps, during which strategic users can guide the algorithm to better align with their true interests by selectively engaging with content. However, users frequently exhibit inconsistent preferences: they may spend considerable time on content that offers little long-term value, inadvertently signaling that such content is desirable. Focusing on the user side, this raises a key question: *what does it take for such users to align the algorithm with their true interests?*

To investigate these dynamics, we model the user's decision process as split between a rational "system 2" that decides whether to engage and an impulsive "system 1" that determines how long engagement lasts. We then study a multi-leader, single-follower extensive Stackelberg game, where users, specifically system 2, lead by committing to engagement strategies and the algorithm best-responds based on observed interactions. We define the burden of alignment as the minimum horizon over which users must optimize to effectively steer the algorithm. We show that a critical horizon exists: users who are sufficiently foresighted can achieve alignment, while those who are not are instead aligned to the algorithm's objective. This critical horizon can be long, imposing a substantial burden. However, even a small, costly signal (e.g., an extra click) can significantly reduce it. Overall, our framework explains how users with inconsistent preferences can align an engagement-driven algorithm with their interests in a Stackelberg equilibrium, highlighting both the challenges and potential remedies for achieving alignment.

## 1 Introduction

We study the interactions between human users and an algorithm over multiple steps. While these interactions may benefit both parties, the user and the algorithm can have misaligned interests. **Our focus is on the user side, asking what it takes for a user to *align* the algorithm with her interests**. We explore this alignment problem in an *interactive* environment where users may exhibit *inconsistent preferences* within an incentive-aware framework. The following explains our setting in detail.

**Our setting and model.** When users have *consistent* preferences—i.e., the engagement length is in proportion to user's true *reward*—the alignment problem reduces to engagement maximization. Recent advances in designing instruction-following language models rely on this assumption of consistent preferences, often using models such as Bradley-Terry [1] to directly or indirectly infer human rewards and optimize them to maximize human approval [2–4]. Similarly, recommender systems are typically designed to optimize recommendations that maximize user engagement [5].

Users can, however, have *inconsistent* preferences, where their actions may not reflect their true interests. This often occurs when a user's decision results from a combination of impulsive system 1 and rational system 2 processes [6–8], or when consumption choices are influenced by both long-term benefits (enrichment) and a desire for instant gratification (temptation) [9]. In such cases, revealed preferences do not necessarily align with the user's true preferences.

39th Conference on Neural Information Processing Systems (NeurIPS 2025).

In general, due to inconsistent preferences or misaligned objectives [10–12], the algorithm's goal may diverge from the user's. Can a strategic user then still succeed in aligning the algorithm with her interests? This question is central to our study, where we examine the conditions and strategies that enable users to steer the algorithm despite such misalignments.

We model the alignment problem as a multi-leader, single-follower Stackelberg game, where users act as leaders by committing to strategies, and the algorithm best responds. We group users with similar interests under the same *user type* to capture preference heterogeneity. Our model reflects real-world settings in which users can initiate or terminate interactions, and excludes scenarios such as strategic classification [13, 14] where the algorithm first commits to a strategy, such as a classifier. By assuming the algorithm best responds, we abstract away its *learning* process [15, 16]. This is reasonable as modern systems interact frequently with users and learn from abundant behavioral data.

Our Stackelberg game captures a dynamic setting where a user interacts with the algorithm over *multiple steps* within a session. After each interaction, the algorithm updates its posterior over the user's type, gradually personalizing its responses. As a result, the strategies of other users affect how well an individual can steer the algorithm. The user's goal is to guide the algorithm toward higher-reward recommendations over time. However, she must trade off long-term signaling with short-term rewards, creating a tension that places the burden of alignment on the user.

**Our results.** To quantify the burden of alignment on the user, we ask: over what future horizon must the user optimize her strategy for the cost of alignment to be worthwhile? We propose this horizon as a measure of burden. When users exhibit inconsistent preferences, system 2 must be sufficiently foresighted to align the algorithm with user's true interests. We show that each user type has a critical threshold: only if the user's horizon exceeds this can she steer the algorithm; otherwise, similar to [17], the algorithm effectively aligns her to its own objective, despite the user leading the game.

The significant burden on users prompts the question: what design features can ease alignment? We explore a setting where users can exert observable effort during each interaction—such as clicking a small, non-beneficial button—as a way to signal system 2's strong disinterest. Though the effort yields no direct reward, it enables users to convey preferences without disengaging, simplifying their strategies. As in the baseline setting, alignment still requires a critical optimization horizon. However, allowing users to "burn effort" as a signal [18] substantially reduces the burden of alignment.

In summary, we present a framework to analyze the alignment problem between an optimal algorithm and users with inconsistent preferences. By fully characterizing the equilibria in a multi-leader Stackelberg game, where users lead, we quantify the burden of alignment in terms of the horizon over which users must optimize. Our framework also guides the design of improved alignment environments by quantifying how costly signaling through effort-burning can reduce this burden. As noted by Dean et al. [19], we believe that such formal mathematical models can provide valuable insights for the broader field of interaction design in future studies.

**Additional related work.** The alignment problem has been studied across computer science, social science, and economics as a game between users and platforms/algorithms. The most closely related setting is a Stackelberg game in which the leader—typically the platform—commits to a strategy over multiple interactions, and the follower—a user—responds. Because recommendations unfold over multiple steps, users may act strategically, anticipating how their responses influence future outcomes. Haupt et al. [20] term such users *strategic*, in contrast to *myopic* users who optimize locally. Similar to our work, they show that users tend to engage in behaviors that accentuate differences relative to users with other preference profiles. Haupt et al. [20] and Cen et al. [21] provide empirical support for this behavior, while Cen et al. [22] study its implications for platform utility.

Our work departs from these studies in two key ways. First, we center users as leaders in the Stackelberg game, focusing on settings where users—particularly system 2—can commit to strategies that the platform observes (e.g., via repeated interactions or prior data). This shift allows us to ask what it takes for users, as leaders, to maximize their own rewards. Second, we extend the analysis to a multi-leader, single-follower game and examine how the presence of other users shapes individual strategies, drawing parallels to Nash equilibrium. Please refer to Appendix A for an extensive review of related work.

## 2 Model

We model the interactions between an *algorithm* and *users*. The users aim to find specific responses or recommendations, while the algorithm seeks to maximize user engagement. We first outline the structure of these interactions and then present a detailed model for both the users and the algorithm.

The user-algorithm interactions occur in *sessions*. Each session involves multiple *interactions* between the algorithm and a *fixed user*. In each interaction, the algorithm suggests an *item/content* and observes whether the user engages with it and for how long. The algorithm uses these observations to refine future suggestions. For users, engagement generates a *reward*, which is higher when the item aligns with their true interest. We assume users have fixed interests throughout a session.

For strategic users, engagement serves a dual purpose: consuming rewards and signaling preferences. If a rational user fully controlled both engagement and its duration, aligning with the algorithm could be as simple as engaging more with preferred content. However, users often exhibit *inconsistent preferences*, sometimes spending time on content misaligned with their true interests. This inconsistency complicates alignment and motivates our study.

### 2.1 Model of human users

We model a user by their *type* $\theta \in \Theta$, which encodes the user's intention during a session. Thus, the same person may have different types across sessions. For simplicity and tractability, we focus on the case where the set of user types $\Theta$ is finite.

Upon receiving a recommendation $s \in S$, the user makes two decisions: whether to engage, and if so, for how long. Let $y \in Y \subseteq \mathbb{R}^{\geq 0}$ denote the user's response, where $y > 0$ indicates engagement lasting $y$ steps. We model human users with potentially inconsistent preferences. The decision to engage is governed by the user's fully rational system 2. If the user chooses to engage, the length of engagement is determined by system 1, with an expected value of $\mathbb{E}[y \mid y > 0; \theta, s] = 1/\alpha_\theta(s)$. Note that in our analysis of optimal decision-making, the expected value of $y$ is the only important factor, which we can interpret as the expected utility of the algorithm when a user of type $\theta$ engages with content $s$. When $y$ follows a geometric distribution with a success rate of $\alpha_\theta(s)$, our model reduces to that of Kleinberg et al. [6].[1]

The user receives an expected reward of $r_\theta(s)$ upon engagement. This reward is independent of engagement length, without loss of generality: even if realized rewards depend on duration, the length is independently governed by system 1, so its effect can be absorbed into the expectation $r_\theta(s)$.

The user discounts future rewards by a factor $\gamma_\mathcal{H} < 1$ per time step.[2] This parameter is central to our analysis of the foresight required for a user to align the algorithm with her interests. We assume a uniform $\gamma_\mathcal{H}$ across all users, though our analysis readily extends to a heterogeneous population.

### 2.2 Model of algorithm

We assume the algorithm maximizes engagement, a common proxy for objectives in human-facing systems; e.g., video recommendation engines optimize for prolonged viewing to increase ad revenue. Our framework also extends beyond engagement: if $y$ is the algorithm's utility from an interaction, the algorithm can be viewed as a utility maximizer. This general perspective encompasses any policy or language model that interacts with users to optimize a utility measure, such as human approval.

Formally, let $H = (s_1, y_1, \cdots, s_T, y_T)$ represent the history of interactions in a session. The algorithm's realized utility from $H$ is $\sum_{t=1}^T y_t$. However, like human users, the algorithm may also be myopic, discounting future returns by a factor of $\gamma_\mathcal{A} < 1$ per step. Thus, the algorithm's valuation of $H$ at the start of the session is $\sum_{t=1}^T \gamma_\mathcal{A}^{t-1} y_t$. Given the models of the users and the algorithm, we next define their respective strategies and introduce a notion of equilibrium.

---

[1]To further clarify the connection to Kleinberg et al. [6], we treat the entire interaction that system 1 has after system 2 decides to engage as a single item. While the user may consume different things during engagement, from system 2's perspective, the only relevant parameter is the expected reward as we will see, and from the platform's perspective, it is the expected length of engagement. What exactly gets consumed during the engagement does not reveal information about the user's type, since it wasn't the result of a rational/strategic decision and the platform only records engagement decisions as we shall see.

[2]We have implicitly assumed that users might discount rewards between interactions, but not during a single interaction, so our previous point on the irrelevance of the length of the engagement and reward remains intact.

# 3 Strategies and equilibria

The algorithm's strategy $\pi : (S \times Y)^* \to S$ maps the session history $H = (s_1, y_1, s_2, y_2, \dots)$ to the next recommendation in general. On the user side, only the engagement decision is strategic: a user of type $\theta$ engages with content $s$ with probability $f_\theta(s)$. We denote the user strategy profile by $\boldsymbol{f} = (f_\theta)_{\theta \in \Theta}$. We now characterize the values induced by these strategies, formulate the optimization problems faced by value-maximizing users and the algorithm, and introduce the resulting equilibria.

## 3.1 Algorithm's strategy and value

For simplicity, we assume that the algorithm only acts on the binary engagement history $\widehat{H} = (s_1, \hat{y}_1 := \mathbb{1}\{y_1 > 0\}, s_2, \hat{y}_2 := \mathbb{1}\{y_2 > 0\}, \dots)$ rather than the full history $H$. Suppose the algorithm has a prior $\boldsymbol{\lambda} = (\lambda_\theta)_{\theta \in \Theta}$ over $\Theta$. The *Q-value* of the algorithm starting with content $s$ is

$$Q_{\mathcal{A}}(\boldsymbol{\lambda}, s; \boldsymbol{f}, \pi) := \mathbb{E}_{\theta \sim \boldsymbol{\lambda}} \Big[ \mathbb{E}_{H \sim (f_\theta, \pi | s)} \Big[ \sum_{(s_t, y_t) \in H} \gamma_{\mathcal{A}}^{t-1} y_t \Big] \Big].$$

Here, $H \sim (f_\theta, \pi \mid s)$ is shorthand for the distribution over the possibly infinite-length history induced by the user strategy $f_\theta$ and the algorithm strategy $\pi$, starting with $s$.

Although the algorithm's policy $\pi$ may depend on the full history $\widehat{H}$, optimal decision-making depends only on the posterior $[\boldsymbol{\lambda} \mid \widehat{H}; \boldsymbol{f}]$ over user types. This follows directly from the Bellman updates, which we prove in Lemma B.1 for completeness. Overloading notation, we define $Q_{\mathcal{A}}(\boldsymbol{\lambda}, s; \boldsymbol{f}) := \max_\pi Q_{\mathcal{A}}(\boldsymbol{\lambda}, s; \boldsymbol{f}, \pi)$. The Bellman update then allows us to compute $Q_{\mathcal{A}}(\boldsymbol{\lambda}, s; \boldsymbol{f})$ recursively as

$$Q_{\mathcal{A}}(\boldsymbol{\lambda}, s; \boldsymbol{f}) = \mathbb{E}_{\theta \sim \boldsymbol{\lambda}} \Big[ \frac{f_\theta(s)}{\alpha_\theta(s)} + \gamma_{\mathcal{A}} \, \mathbb{E}_{\hat{y} \sim \text{Ber}(f_\theta(s))} \Big[ \max_{s'} Q_{\mathcal{A}} \big( [\boldsymbol{\lambda} \mid (s, \hat{y}); \boldsymbol{f}], s'; \boldsymbol{f} \big) \Big] \Big]. \quad (1)$$

This plays a pivotal role in our analysis of the best strategies. Once $Q_{\mathcal{A}}$ is found, the optimal policy is

$$\pi^*(\widehat{H}) \in \arg\max_s Q_{\mathcal{A}} \big( [\boldsymbol{\lambda} \mid \widehat{H}; \boldsymbol{f}], s; \boldsymbol{f} \big).$$

## 3.2 User's strategy and value

The user's strategy specifies the engagement probability $f_\theta : S \to [0, 1]$ over content space $S$. Given the algorithm's strategy $\pi$ and initial content $s$, the *Q-value* for a user of type $\theta$ is

$$Q_\theta(s; \boldsymbol{f}, \pi) := \mathbb{E}_{H \sim (f_\theta, \pi | s)} \Big[ \sum_{(s_t, y_t) \in H} \mathbb{1}\{y_t > 0\} \gamma_{\mathcal{H}}^{t-1} \, r_\theta(s_t) \Big].$$

Let $\boldsymbol{\lambda}$ denote the algorithm's current posterior over user types. Suppose the algorithm uses the strategy profile $\boldsymbol{f}'$ to calculate this posterior, where $\boldsymbol{f}'$ is not necessarily the same as $\boldsymbol{f}$. Overloading the notation, we obtain the following Bellman update for the user's Q-value:

$$\begin{aligned} Q_\theta(\boldsymbol{\lambda}, s; \boldsymbol{f}, \boldsymbol{f}') = {} & f_\theta(s) \, r_\theta(s) \\ & + \gamma_{\mathcal{H}} \, \mathbb{E}_{\hat{y} \sim \text{Ber}(f_\theta(s))} Q_\theta \Big( [\boldsymbol{\lambda} \mid (s, \hat{y}); \boldsymbol{f}'], \arg\max_{s'} Q_{\mathcal{A}} \big( [\boldsymbol{\lambda} \mid (s, \hat{y}); \boldsymbol{f}'], s'; \boldsymbol{f}' \big); \boldsymbol{f}, \boldsymbol{f}' \Big). \end{aligned} \quad (2)$$

When taking the $\arg\max$, we can assume any item that maximizes the $Q_{\mathcal{A}}$ is chosen.

## 3.3 Equilibrium definition

We consider a multi-leader, single-follower Stackelberg equilibrium. In this setup, users (leaders) commit to a strategy $\boldsymbol{f}$, and the algorithm (follower) best responds to $\boldsymbol{f}' = \boldsymbol{f}$. Although the algorithm does not directly observe $\boldsymbol{f}$, the assumption of $\boldsymbol{f}' = \boldsymbol{f}$ is reasonable given that modern algorithms have access to vast amounts of data and computational resources to infer user strategies. Additionally, we assume that no individual user (leader) has an incentive to unilaterally deviate from the equilibrium strategy. From the users' perspective, this implies they are in a mixed-strategy Nash equilibrium.

To formalize equilibrium, we consider two session entry scenarios: *random entry* (RE) and *algorithmic entry* (AE). Under random entry, the user stumbles upon an initial content $s$, for example

by landing on a video platform. Let $s$ be drawn from a distribution $p_1$, and let the prior over user types $\boldsymbol{\lambda}$ be common knowledge. At equilibrium, for every $\theta \in \Theta$, we have

$$f_\theta^{\text{RE}} \in \arg\max_{f_\theta} V_\theta^{\text{RE}}\big(\boldsymbol{\lambda}; (f_\theta, \boldsymbol{f}_{-\theta}^{\text{RE}})\big) \coloneqq \mathbb{E}_{s_1 \sim p_1}\Big[Q_\theta\big(\boldsymbol{\lambda}, s_1; (f_\theta, \boldsymbol{f}_{-\theta}^{\text{RE}}), (f_\theta, \boldsymbol{f}_{-\theta}^{\text{RE}})\big)\Big]. \quad (3)$$

In the case of *algorithmic entry*, the algorithm recommends the first item to the user at the start of the session. Given that the prior over user types $\boldsymbol{\lambda}$ is common knowledge, the recommended item is $s_1 \in \arg\max_s Q_\mathcal{A}(\boldsymbol{\lambda}, s; \boldsymbol{f})$. Thus, at equilibrium, for every $\theta \in \Theta$, we have

$$f_\theta^{\text{AE}} \in \arg\max_{f_\theta} V_\theta^{\text{AE}}\big(\boldsymbol{\lambda}; (f_\theta, \boldsymbol{f}_{-\theta}^{\text{AE}})\big) \coloneqq Q_\theta\Big(\boldsymbol{\lambda}, \arg\max_s Q_\mathcal{A}\big(\boldsymbol{\lambda}, s; (f_\theta, \boldsymbol{f}_{-\theta}^{\text{AE}})\big); (f_\theta, \boldsymbol{f}_{-\theta}^{\text{AE}}), (f_\theta, \boldsymbol{f}_{-\theta}^{\text{AE}})\Big). \quad (4)$$

The study of these general notions of equilibrium can be intractable. Therefore, we next define a special case of interest that enables us to characterize equilibria and analyze their properties.

### 3.4 Special case: Inconsistent actions and rewards

For a user of type $\theta$, steering the algorithm via system 2's engagement decisions is challenging when system 1's engagement length (or, equivalently, the algorithm's utility) is misaligned with the user's reward. This challenge is amplified when users with complementary interests shape the algorithm's default behavior, making it harder for type $\theta$ to distinguish herself. We formalize this case below.

Suppose there are two possible (types of) items: $S = \{a, b\}$. Item $a$ is more tempting for everyone, so $1/\alpha_\theta(a) > 1/\alpha_\theta(b)$ for every $\theta \in \Theta$. For some types of users, $\Theta_1 \subseteq \Theta$, item $b$ is more rewarding, i.e., $r_\theta(b) > r_\theta(a)$, however, the remaining types $\Theta_2 = \Theta \setminus \Theta_1$ have interests aligned with the algorithm, i.e., $r_\theta(a) > r_\theta(b)$. We refer to users with type $\theta \in \Theta_1$ as type 1 users and those with type $\theta \in \Theta_2$ as type 2 users. For easy reference, we summarize this special case in Table 1. Note that type $a$ and $b$ contents can have different interpretations, such as being popular versus niche items [5].

Table 1: Special case of interest where type 1 users have inconsistent actions and rewards

| User type | Engagement length | Reward |
|---|---|---|
| $\theta \in \Theta_1$ | $\frac{1}{\alpha_\theta(a)} > \frac{1}{\alpha_\theta(b)}$ | $r_\theta(a) < r_\theta(b)$ |
| $\theta \in \Theta_2$ | | $r_\theta(a) > r_\theta(b)$ |

When type 1 users engage with content $a$, the algorithm receives utility $1/\alpha_\theta(a)$—higher than from content $b$. If system 2 always engages, the algorithm has no incentive to recommend $b$. To steer the algorithm away from $a$, system 2 must reduce its engagement probability such that $f_\theta(a) < \frac{\alpha_\theta(a)}{\alpha_\theta(b)}$. Refusing to engage (1) signals the user's type, (2) discourages recommendation of tempting content, but (3) incurs a cost of $(1 - f_\theta(a)) r_\theta(a)$ when $a$ is shown. This trade-off complicates the user's strategy. In the next section, we will analyze the resulting best strategies of this special case, but before that, to further contextualize our model and this special case, consider two examples:

*Example* 1. A user opens a music recommender system while working, with the intent of listening to calm music. If the user (system 2) chooses to engage with the platform, they select a starting music $s$, after which the platform autoplays subsequent items. The number of musics autoplayed after $s$ until the user disengages and selects another entry music $s'$ defines the length of engagement $y$, determined by system 1. The platform benefits from longer engagement, e.g., through ad revenue, whereas the user benefits from listening to calm music while working. However, suppose the user is also a fan of singer X, whose music is engaging but distracting. This user is then a type 1 user, and the platform must choose between recommending calm music (type $b$) or X's music (type $a$) during the working session. Note that during the working session, the user's intent remains fixed. Yet, on a later occasion, say, when relaxing, the same user's intent may shift toward listening to X's music for an extended period, effectively becoming a type 2 user.

*Example* 2. A chatbot that charges per API call may operate in several "modes." In an educational mode, longer conversations are valuable for a student, aligning incentives between user and platform (type 2). In contrast, for an engineer seeking a quick answer, shorter sessions are preferable (type 1). Similarly, a therapy chatbot operating in an affirmative mode may sustain longer conversations by offering emotionally validating responses, even if doing so delays meaningful progress. Here, deciding whether to engage with a psychologist chatbot in a mode is a system 2 decision, while the duration of the conversation is governed by system 1, and what should be the default mode every time while the user is going through weeks of therapy is the algorithm's choice.

## 4 Characterizing equilibria: Algorithm's best response

In this section, we characterize the algorithm's optimal strategy for the special case outlined in Sec. 3.4. Specifically, we show that, under reasonable assumptions, the algorithm's Q-value is piecewise linear in the prior $\boldsymbol{\lambda}$ over user types and that the algorithm's strategy behaves as a linear classifier acting on $\boldsymbol{\lambda}$. Building on this, in the next section, we will characterize the users' optimal strategies for maximizing their value at equilibrium under both random and algorithmic entries.

To solve the algorithm's Q-value from the Bellman update in Eq. (1), we first restrict $\boldsymbol{f}$ under the following reasonable assumptions to avoid pathological cases.

**Assumption 1.** *Let $s_\theta^* \in \arg\max_s r_\theta(s)$ be the highest rewarding content for user type $\theta$. We assume that every user of type $\theta$ always engages with $s_\theta^*$, i.e., $f_\theta(s_\theta^*) = 1, \forall \theta \in \Theta$, and no user chooses occasional engagement with $s$, i.e., $f_\theta(s) < 1$, if that does not discourage the algorithm about $s$, so*

$$f_\theta(s) \in \left[0, \frac{\alpha_\theta(s)}{\alpha_\theta(s_\theta^*)}\right) \cup \{1\}, \qquad \forall \theta \in \Theta, \forall s \in S.$$

In our special case of interest, this assumption implies

$$f_\theta(a) \in \left[0, \frac{\alpha_\theta(a)}{\alpha_\theta(b)}\right) \cup \{1\}, \quad f_\theta(b) = 1, \qquad \forall \theta \in \Theta_1, \tag{5}$$

$$f_\theta(a) = 1, \qquad\qquad\qquad\qquad \forall \theta \in \Theta_2. \tag{6}$$

Given this restricted user strategy profile, we now present the algorithm's best response:

**Theorem 4.1** (Algorithm's best response)**.** *Given that the algorithm has a posterior $\boldsymbol{\lambda}$ over $\Theta$, it will best respond by recommending item $a$ if and only if $\sum_{\theta \in \Theta} h_\theta \lambda_\theta \geq 0$, where*

$$h_\theta = \frac{1 - \gamma_\mathcal{A}}{(1 - \gamma_\mathcal{A} f_\theta(a)) (1 - \gamma_\mathcal{A} f_\theta(b))} \left[\frac{f_\theta(a)}{\alpha_\theta(a)} - \frac{f_\theta(b)}{\alpha_\theta(b)}\right]. \tag{7}$$

See proof on page 22. This theorem shows that the algorithm best responds by using a linear classifier $\boldsymbol{h} = (h_\theta)_{\theta \in \Theta}$, which acts on the posterior $\boldsymbol{\lambda}$. Interestingly, $h_\theta$ depends only on the strategy of user type $\theta$, i.e., $f_\theta$, with no interaction between users appearing in $\boldsymbol{h}$. This simplifies our characterization of the equilibrium, as we will discuss next.

## 5 Characterizing equilibria: User's best response

Given the algorithm's best response, we now analyze the user's best response, which defines the Stackelberg equilibrium of our game. We demonstrate that the user's strategy often takes a simple form that allows us to identify all equilibria. We then examine the user's regret under equilibrium over a finite horizon. This regret is measured using an undiscounted sum of realized rewards, reflecting how much reward the user forfeits due to being myopic (i.e., having $\gamma_\mathcal{H} < 1$). We show that a user incurs a constant regret only if they are sufficiently foresighted, meaning $\gamma_\mathcal{H}$ exceeds a threshold specific to their type.

Starting from Theorem 4.1, we observe that the algorithm uses a linear classifier to make its decisions. Here, a user of type $\theta$ contributes to the classifier's margin by an amount of $h_\theta \lambda_\theta$. Therefore, this user can influence the classifier to recommend item $a$ (or item $b$) by choosing a larger (or smaller) value of $h_\theta$. For instance, a type 1 user has $h_\theta \propto f_\theta(a) - \alpha_\theta(a)/\alpha_\theta(b)$. This user can push the classifier's value toward a negative value, favoring item $b$, by refusing to engage, i.e., setting $f_\theta(a)$ below $\alpha_\theta(a)/\alpha_\theta(b)$.

A type $\theta$ user's ability to steer the algorithm toward her preferred item depends on the strategies of other users—specifically, on the *classifier margin from the perspective of type $\theta$*:

$$m_\theta := \sum_{\theta' \in \Theta \setminus \{\theta\}} h_{\theta'} \lambda_{\theta'}. \tag{8}$$

Intuitively, a larger margin implies that a type 1 user will have greater difficulty steering the algorithm. We now formalize this intuition by fully characterizing the equilibria under algorithmic entry:

**Theorem 5.1** (Equilibrium under algorithmic entry). *Let $m_\theta^{\mathrm{AE}}$ be the margin of the algorithm's classifier from the perspective of user type $\theta$ when all other user types follow the equilibrium strategy under algorithmic entry. Define the* steerable sets *for type 1 and 2 users as follows:*

$$\theta \in \Theta_1 : \ F_\theta := \left\{ x \in \left[0, \frac{\alpha_\theta(a)}{\alpha_\theta(b)}\right) \mid \frac{\lambda_\theta}{\alpha_\theta(b)} - m_\theta^{\mathrm{AE}} - x \left(\frac{\lambda_\theta}{\alpha_\theta(a)} - \gamma_{\mathcal{A}} \, m_\theta^{\mathrm{AE}}\right) \geq 0 \right\}$$

$$\theta \in \Theta_2 : \ F_\theta := \left\{ x \in [0, 1] \mid \frac{\lambda_\theta}{\alpha_\theta(a)} + m_\theta^{\mathrm{AE}} - x \left(\frac{\lambda_\theta}{\alpha_\theta(b)} + \gamma_{\mathcal{A}} \, m_\theta^{\mathrm{AE}}\right) \geq 0 \right\}.$$

*Let $s_\theta^*$ and $(-s_\theta^*)$ be the high and low reward contents for type $\theta$. The user's strategy at equilibrium is*

$$f_\theta^{\mathrm{AE}}(s_\theta^*) = 1, \quad f_\theta^{\mathrm{AE}}(-s_\theta^*) = \begin{cases} \text{any value in } F_\theta, & F_\theta \neq \emptyset, \\ 0, & F_\theta = \emptyset, \gamma_{\mathcal{H}} > \frac{r_\theta(-s_\theta^*)}{r_\theta(s_\theta^*)}, \\ 1, & F_\theta = \emptyset, \gamma_{\mathcal{H}} < \frac{r_\theta(-s_\theta^*)}{r_\theta(s_\theta^*)}, \\ \text{any value in } [0,1], & F_\theta = \emptyset, \gamma_{\mathcal{H}} = \frac{r_\theta(-s_\theta^*)}{r_\theta(s_\theta^*)}. \end{cases}$$

See proof on page 24. This theorem shows that for each user type $\theta$, there exists a steerable set $F_\theta$, defined by a linear constraint on the user's strategy. If nonempty, any strategy in $F_\theta$ results in an equilibrium. Notably, a larger margin $m_\theta^{\mathrm{AE}}$ shrinks $F_\theta$ for type 1 users and expands it for type 2.

The definition of $F_\theta$ also highlights the role of $\gamma_{\mathcal{A}}$, which captures the algorithm's foresight. Notably, $\gamma_{\mathcal{A}}$ appears only in the product $\gamma_{\mathcal{A}} \, m_\theta^{\mathrm{AE}}$, so its effect depends on both the sign and magnitude of the margin. For instance, when $m_\theta^{\mathrm{AE}} > 0$, increasing $\gamma_{\mathcal{A}}$ expands the steerable set for type 1 users: as the algorithm becomes more foresighted, even slight disengagement from type 1 users can effectively influence its behavior. Moreover, the steerable set for type 1 users has the following structure:

**Corollary 5.2.** *The steerable set for user type $\theta \in \Theta_1$ is nonempty if and only if $\lambda_\theta \geq \alpha_\theta(b) \, m_\theta^{\mathrm{AE}}$. Moreover, when nonempty, $F_\theta = [0, c)$ for some $c$.*

See proof on page 25. As an immediate result of this corollary, we have the following observation:

**Corollary 5.3.** *For a user of type $\theta \in \Theta_1$, in any equilibrium under algorithmic entry where $m_\theta^{\mathrm{AE}} > \lambda_\theta / \alpha_\theta(b)$ and $\gamma_{\mathcal{H}} \neq r_\theta(a) / r_\theta(b)$, the user's strategy is*

$$f_\theta^{\mathrm{AE}}(b) = 1, \quad f_\theta^{\mathrm{AE}}(a) = \mathbb{1}\left\{\gamma_{\mathcal{H}} < \frac{r_\theta(a)}{r_\theta(b)}\right\}.$$

This observation extends beyond algorithmic entry: in Theorem B.2, deferred to the appendix for conciseness, we show that the same user strategy also holds at equilibrium under random entry.

When the steerable set is empty, these results imply that—regardless of whether entry is algorithmic or random—a strategic user will disengage from undesired content only if she is sufficiently foresighted. Let $\tau_{\mathcal{H}} := 1/(1 - \gamma_{\mathcal{H}})$ be the user's *effective horizon*. Type 1 users disengage only if

$$\tau_{\mathcal{H}} := \frac{1}{1 - \gamma_{\mathcal{H}}} > \frac{r_\theta(b)}{r_\theta(b) - r_\theta(a)}. \tag{9}$$

If the effective horizon is short, the user will fully engage with the tempting content, aligning with the *algorithm's interest*. This can lead to significant regret for the user, as we shall discuss next.

**Regret under equilibrium strategies.** While we used $\gamma_{\mathcal{H}}$-discounted rewards to model the user's limited foresight, comparing strategies requires an undiscounted sum of rewards. Let $V_\theta^T(\boldsymbol{\lambda}; \boldsymbol{f})$ denote the expected total reward over $T$ steps for a user of type $\theta$, given a strategy profile $\boldsymbol{f}$, a type distribution $\boldsymbol{\lambda}$, and an algorithm that best responds:

$$V_\theta^T(\boldsymbol{f}) := \mathbb{E}_{H_T \sim (f_\theta, \pi^*)} \left[ \sum_{(s_t, y_t) \in H_T} \mathbb{1}\{y_t > 0\} \, r_\theta(s_t) \right].$$

Here, $H_T \sim (f_\theta, \pi^*)$ denotes the distribution of histories of length $T$ when the user leads by the strategy $f_\theta$ and the algorithm best responds. Note that the first item in the history may be either randomly chosen or optimally selected by the algorithm. We dropped $\boldsymbol{\lambda}$ from the notation for brevity.

Let $\boldsymbol{f}^{\text{xE},\gamma_{\mathcal{H}}\to 1}$ be the best user strategy profile under x-entry (either random or algorithmic) when $\gamma_{\mathcal{H}}\to 1$. We define the regret of user type $\theta$ after $T$ steps as

$$\text{Regret}_\theta^T(\boldsymbol{f}^{\text{xE}}) := V_\theta^T(\boldsymbol{f}^{\text{xE},\gamma_{\mathcal{H}}\to 1}) - V_\theta^T(\boldsymbol{f}^{\text{xE}}).$$

We can see that a user incurs constant regret—spending only a fixed number of time steps aligning the algorithm with her preferred content—if and only if her effective horizon is sufficiently large:

**Corollary 5.4.** *A user of type $\theta \in \Theta_1$ has constant regret in equilibrium if and only if Eq. (9) holds.*

See proof on page 25.

# 6 Costly signaling reduces the burden of alignment

In this section, we show that if the platform enables costly signaling—such as allowing the user to click a small button to show disinterest—then system 2 can incur this additional cost to better signal its type. The possibility of incurring a cost effectively separates type communication from content consumption and can ease the user's engagement decisions.

We begin by formalizing costly signaling and redefining the user's strategy and equilibrium. As before, we first derive the algorithm's best response, then characterize the user's. Finally, we quantify how much introducing costly signaling can alleviate the user's burden to achieve constant regret.

## 6.1 Strategies, values, and equilibrium

We assume that the user's system 2 can choose to incur a fixed cost $c$, and this effort is observable by the algorithm. A user of type $\theta$ adopts a strategy that, when presented with item $s$, involves both the probability of engagement, denoted by $f_\theta(s)$, and the probability of incurring the cost, denoted by $u_\theta(s)$. We denote the strategy profile of the users by $(\boldsymbol{f}, \boldsymbol{u})$, where $\boldsymbol{u} := (u_\theta)_{\theta\in\Theta}$.

The algorithm observes a history of both engagements and costs: $\widehat{H} = (s_1, \hat{y}_1, \hat{u}_1, s_2, \hat{y}_2, \hat{u}_2, \cdots)$, where $\hat{u}$ is a binary indicator of whether the cost was incurred. As before, the algorithm maintains a posterior distribution over user types given $\widehat{H}$ to best respond, which yields the Bellman update

$$Q_{\mathcal{A}}(\boldsymbol{\lambda}, s; \boldsymbol{f}, \boldsymbol{u}) = \mathbb{E}_{\theta\sim\boldsymbol{\lambda}}\left[\frac{f_\theta(s)}{\alpha_\theta(s)} + \gamma_{\mathcal{A}} \mathbb{E}_{\substack{\hat{y}\sim\text{Ber}(f_\theta(s)) \\ \hat{u}\sim\text{Ber}(u_\theta(s))}}\left[\max_{s'} Q_{\mathcal{A}}\left([\boldsymbol{\lambda} \mid (s,\hat{y},\hat{u}); \boldsymbol{f}, \boldsymbol{u}], s'; \boldsymbol{f}, \boldsymbol{u}\right)\right]\right]. \quad (10)$$

Let $\boldsymbol{\lambda}$ denote the current algorithm's posterior over user types. Suppose the algorithm uses the strategy profile $(\boldsymbol{f}', \boldsymbol{u}')$ to calculate this posterior, where $\boldsymbol{f}'$ and $\boldsymbol{u}'$ are not necessarily the same as $\boldsymbol{f}$ and $\boldsymbol{u}$. We have the following Bellman update for the user type $\theta$'s Q-value:

$$Q_\theta(\boldsymbol{\lambda}, s; \boldsymbol{f}, \boldsymbol{u}, \boldsymbol{f}', \boldsymbol{u}') = f_\theta(s)\, r_\theta(s) - u_\theta(s)\, c \quad (11)$$

$$+ \gamma_{\mathcal{H}} \mathbb{E}_{\substack{\hat{y}\sim\text{Ber}(f_\theta(s)) \\ \hat{u}\sim\text{Ber}(u_\theta(s))}} Q_\theta\left([\boldsymbol{\lambda} \mid (s,\hat{y},\hat{u}); \boldsymbol{f}', \boldsymbol{u}'], \arg\max_{s'} Q_{\mathcal{A}}([\boldsymbol{\lambda} \mid (s,\hat{y},\hat{u}); \boldsymbol{f}', \boldsymbol{u}'], s'; \boldsymbol{f}', \boldsymbol{u}'); \boldsymbol{f}, \boldsymbol{u}, \boldsymbol{f}', \boldsymbol{u}'\right).$$

We study a multi-leader, single-follower Stackelberg equilibrium where users (leaders) commit to a strategy $(\boldsymbol{f}, \boldsymbol{u})$, and the algorithm (follower) best responds to $(\boldsymbol{f}' = \boldsymbol{f}, \boldsymbol{u}' = \boldsymbol{u})$. For brevity, we omit $(\boldsymbol{f}', \boldsymbol{u}')$ from $Q_\theta$ notation and focus on the case of algorithmic entry. At equilibrium, we have

$$(f_\theta^{\text{AE}}, u_\theta^{\text{AE}}) \in \arg\max_{f_\theta, u_\theta} Q_\theta\left(\boldsymbol{\lambda}, \arg\max_s Q_{\mathcal{A}}\left(\boldsymbol{\lambda}, s; (f_\theta, \boldsymbol{f}_{-\theta}^{\text{AE}}), (u_\theta, \boldsymbol{u}_{-\theta}^{\text{AE}})\right); (f_\theta, \boldsymbol{f}_{-\theta}^{\text{AE}}), (u_\theta, \boldsymbol{u}_{-\theta}^{\text{AE}})\right). \quad (12)$$

We next characterize equilibria of this form and show they impose a lower burden on the user.

## 6.2 Characterizing equilibria: Algorithm's best response

We characterize the algorithm's optimal strategy, focusing on the special case outlined in Sec. 3.4. We show that, similar to the case without signaling, the algorithm's Q-value is piecewise linear in the prior $\boldsymbol{\lambda}$ over user types, and the algorithm's strategy functions as a linear classifier operating on $\boldsymbol{\lambda}$. To solve the Bellman update in Eq. (10), we impose an additional restriction on $(\boldsymbol{f}, \boldsymbol{u})$ beyond Assumption 1 to rule out pathological cases:

**Assumption 2.** *Let $s_\theta^* \in \arg\max_s r_\theta(s)$ be the highest rewarding content for user type $\theta$. We assume that no user of type $\theta$ pays a cost when recommended with $s_\theta^*$, i.e., $u_\theta(s_\theta^*) = 0$, $\forall \theta \in \Theta$, and no user pays a cost for content $s$ if that does not discourages the algorithm from recommending $s$:*

$$u_\theta(s) > 0 \implies f_\theta(s) \in \left[0, \frac{\alpha_\theta(s)}{\alpha_\theta(s_\theta^*)}\right), \qquad \forall \theta \in \Theta, \forall s \in S.$$

Given these restrictions over the user strategy profile, we now present the algorithm's best response:

**Theorem 6.1** (Algorithm's best response with signaling). *Given that the algorithm has a posterior $\boldsymbol{\lambda}$ over $\Theta$, it will best respond by recommending item $a$ if and only if $\sum_{\theta \in \Theta} h_\theta \lambda_\theta \geq 0$, where*

$$h_\theta = \frac{1 - \gamma_{\mathcal{A}}}{\left(1 - \gamma_{\mathcal{A}} f_\theta(a)(1 - u_\theta(a))\right)\left(1 - \gamma_{\mathcal{A}} f_\theta(b)(1 - u_\theta(b))\right)} \left[\frac{f_\theta(a)}{\alpha_\theta(a)} - \frac{f_\theta(b)}{\alpha_\theta(b)}\right]. \quad (13)$$

See proof on page 25. We next discuss the characterization of the equilibrium with signaling.

### 6.3 Characterizing equilibria: User's best response

Given the algorithm's best response, we now analyze the user's best, we now formally characterize the equilibria under algorithmic entry when users can incur an observable cost $c$:

**Theorem 6.2** (Equilibrium under algorithmic entry with signaling). *Let $m_\theta^{\mathrm{AE}}$ be the margin of the algorithm's classifier from the perspective of user type $\theta$ when other user types follow the equilibrium strategy under algorithmic entry with signaling. Define the* steerable sets *for type 1 and 2 users as*

$$\theta \in \Theta_1: \ F_\theta := \left\{(x, y) \in \left[0, \frac{\alpha_\theta(a)}{\alpha_\theta(b)}\right) \times [0, 1] \ \Big| \ \frac{\lambda_\theta}{\alpha_\theta(b)} - m_\theta^{\mathrm{AE}} - x\left(\frac{\lambda_\theta}{\alpha_\theta(a)} - \gamma_{\mathcal{A}}(1 - y) m_\theta^{\mathrm{AE}}\right) \geq 0\right\}$$

$$\theta \in \Theta_2: \ F_\theta := \left\{(x, y) \in [0, 1]^2 \ \Big| \ \frac{\lambda_\theta}{\alpha_\theta(a)} + m_\theta^{\mathrm{AE}} - x\left(\frac{\lambda_\theta}{\alpha_\theta(b)} + \gamma_{\mathcal{A}}(1 - y) m_\theta^{\mathrm{AE}}\right) \geq 0\right\}.$$

*Let $s_\theta^*$ and $(-s_\theta^*)$ be the high and low reward contents for type $\theta$. Define the critical $\gamma_{\mathcal{H}}$ value for type $\theta$ as*

$$\gamma_{\mathcal{H}}^c := \frac{c + r_\theta(-s_\theta^*)\left(1 - \frac{\alpha_\theta(-s_\theta^*)}{\alpha_\theta(s_\theta^*)}\right)}{c + r_\theta(s_\theta^*) - r_\theta(-s_\theta^*)\frac{\alpha_\theta(-s_\theta^*)}{\alpha_\theta(s_\theta^*)}} \ .$$

*Assume $\gamma_{\mathcal{H}} \neq \gamma_{\mathcal{H}}^c$ and $c < \frac{\alpha_\theta(-s_\theta^*)}{\alpha_\theta(s_\theta^*)} r_\theta(-s_\theta^*)$. The user's strategy at equilibrium is*

$$(f_\theta^{\mathrm{AE}}(s_\theta^*), u_\theta^{\mathrm{AE}}(s_\theta^*)) = (1, 0),$$

$$(f_\theta^{\mathrm{AE}}(-s_\theta^*), u_\theta^{\mathrm{AE}}(-s_\theta^*)) = \begin{cases} \text{any value in } F_\theta, & F_\theta \neq \emptyset, \\ \left(\frac{\alpha_\theta(-s_\theta^*)}{\alpha_\theta(s_\theta^*)}, 1\right), & F_\theta = \emptyset, \gamma_{\mathcal{H}} > \gamma_{\mathcal{H}}^c, \\ (1, 0), & F_\theta = \emptyset, \gamma_{\mathcal{H}} < \gamma_{\mathcal{H}}^c. \end{cases}$$

See proof on page 27. When signaling via incurred costs is allowed, the steerable set is defined by bilinear constraints over the user strategy. Several key insights follow. First, compared to Theorem 5.1, the projection of the steerable set onto the $f_\theta$ dimension is at least as large with costly signaling as without. Second, for both user types, incurring a cost can expand the steerable set when the margin works against them. Specifically, for type 1 (type 2) users with $m_\theta^{\mathrm{AE}} > 0$ ($m_\theta^{\mathrm{AE}} < 0$), if $(x, y) \in F_\theta$, then so is any $(x, y')$ with $y' \geq y$. Finally, as in the no-signaling case, the steerable set for type 1 users is nonempty if and only if $\lambda_\theta \geq \alpha_\theta(b), m_\theta^{\mathrm{AE}}$. The proof parallels the argument in Corollary 5.2.

When the steerable set is empty and the signaling cost is sufficiently small, Theorem 6.2 contrasts sharply with Theorem 5.1. In this case, sufficiently foresighted users optimally choose to incur the cost and *partially* engage with undesired content. This allows them to decouple type signaling from reward consumption: they fully communicate their type by paying the cost, while limiting engagement. Formally, for type 1 users, we state the following result under an empty steerable set:

**Corollary 6.3.** *For a user of type $\theta \in \Theta_1$, in any equilibrium under algorithmic entry with costly signaling where $m_\theta^{\mathrm{AE}} > \lambda_\theta/\alpha_\theta(b)$ and the signaling cost is sufficiently small, the user's strategy is*

$$f_\theta^{\mathrm{AE}}(b) = 1, \ u_\theta^{\mathrm{AE}}(b) = 0,$$

$$f_\theta^{\mathrm{AE}}(a) \to \mathbb{1}\{\gamma_{\mathcal{H}} < \gamma_{\mathcal{H}}^c\} \cdot \frac{\alpha_\theta(a)}{\alpha_\theta(b)}, \ u_\theta^{\mathrm{AE}}(a) = 1.$$

As an immediate result, one can verify that, similar to the case without signaling, the user must still be sufficiently foresighted to achieve constant regret in every case:

**Corollary 6.4.** *Assuming the cost of signaling is sufficiently small, a user of type $\theta \in \Theta_1$ will always have constant regret in equilibrium under algorithmic entry with signaling if and only if*

$$\tau_{\mathcal{H}} > \frac{1}{1 - \gamma_{\mathcal{H}}^c} = \frac{r_\theta(b) - \left(r_\theta(a)\frac{\alpha_\theta(a)}{\alpha_\theta(b)} - c\right)}{r_\theta(b) - r_\theta(a)} \, .$$

Compared to Corollary 5.4, a key insight emerges: costly signaling lowers the burden of alignment. Specifically, the required effective horizon for a type 1 user is relatively reduced by

$$\frac{r_\theta(a)}{r_\theta(b)} \frac{\alpha_\theta(a)}{\alpha_\theta(b)} - \frac{c}{r_\theta(b)} \, .$$

In conclusion, the opportunity to signal by incurring a cost can both expand the steerable set for users and reduce the alignment burden, requiring optimization over a shorter horizon.

# 7 Discussion

We presented a formal framework to examine the burden of alignment in settings where users have inconsistent preferences. Given the vast array of design choices in such contexts, mathematical modeling is essential to understand the trade-offs and inform practice about the limitations of alignment and potential solutions [19, 23].

Our analysis assumes the platform seeks to maximize engagement or utility—a reasonable goal for self-interested platforms that benefit from user interaction. However, one way to reduce the burden of alignment is to reconsider this objective, if modifiable. Alternatives include optimizing for long-term returns [8], user enrichment [9], or societal objectives [24]. These approaches often require inferring user mental states that are not directly observable, but must be inferred from behavioral data [25].

While our study focuses on misalignment between user and algorithmic interests, it is important to note that engagement maximization may still produce unintended outcomes even without explicit misalignment. For example, differences in user feedback rates across content types can inadvertently lead the algorithm to favor certain types of content [26].

Our work has several limitations. We simplify the learning process by assuming that users are aware of their rewards and that the algorithm has full knowledge of strategies. Additionally, we focus on a two-sided interaction between the platform and users, while real-world scenarios often include a third side—content creators—who may strategically invest in different content types [27].

In summary, our work highlights a critical challenge in alignment from the user's perspective. By providing an economic framework, we contribute to a deeper understanding of the limitations of alignment and the importance of modeling human decision-making. This framework can also inform human–computer interaction design [28, 29] on how to better accommodate diverse user preferences and behavior.

## Acknowledgments

This project began through formative research conversations with Moritz Hardt. It later took a new direction following the author's discussion with Jason Hartline at FORC'23. The author thanks Dylan Hadfield-Menell for extensive feedback on an earlier version of this work. The author also thanks Jon Kleinberg, as well as participants of the EC'25 Workshop on Information Economics x Large Language Models and the EC'25 Workshop on Swap Regret and Strategic Learning, for their helpful feedback on a poster version of this work. The author is further grateful to the anonymous reviewers for their thoughtful comments and deep engagement with the paper.

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

# A   Extensive review of related work

The alignment problem has been studied as a game between users and platforms/algorithms across computer science, social science, and economics. In this section, we provide an overview of the work most relevant to our study.

Much of the literature models user–algorithm interactions as Stackelberg games in various forms. In one common variation, the algorithm leads by choosing a recommendation policy while users myopically best respond by selecting the highest-rewarding option. In this setup, engagement maximization may yield highly suboptimal outcomes for users [5], and when the algorithm employs an online learning procedure, its sample complexity for regret minimization can be exponential [30].

A setting closer to ours features Stackelberg games where the leader (typically a platform) commits to an extensive strategy over multiple interactions, and the follower (typically a user) best responds. Because recommendations occur over several steps, users account for how their response to each item affects future interactions. Haupt et al. [20] call these users *strategic users*—in contrast to *myopic users* who optimize locally. Similar to our work, they show that users (in particular, minorities) tend to engage in behaviors that accentuate differences relative to users with other preference profiles. Both Haupt et al. [20] and Cen et al. [21] provide empirical evidence from lab and online experiments supporting such strategization. Similarly, Cen et al. [22] analyze a two-player game focusing on how these behaviors help or hurt the platform in the short and long term, while Hébert and Zhong [31] demonstrate that when the principal maximizes engagement, the agent may, in the worst case, fail to extract any useful information or utility from her interactions.

Our work diverges from these studies in two important ways. First, we treat users as the leaders in the Stackelberg game. We assume that users—particularly system 2—can commit to an engagement strategy, and the platform or algorithm observes this commitment (for example, via repeated interactions and prior data). Consequently, we focus on what it takes for users, as leaders, to maximize their reward when engaging with the algorithm. Second, our analysis generalizes to a multi-leader single-follower game, and we study how the presence of other users affects each individual's strategy in a manner similar to Nash equilibrium.

Strategic classification [13, 14] offers another related perspective. In that setting, the platform typically publishes a classifier and users strategically respond—often incurring a cost of change—to optimize their outcomes. In contrast, our framework features users who first commit to an engagement strategy. Moreover, while strategic classification typically unfolds at a single time point, we model extensive-form games in which the follower's strategy depends on the history of interactions. At a high level, both settings found the relative capabilities of the parties—in our case, on how foresighted the users—determine who is following and who is leading [17].

We work in a full-information setting, which means (1) users know their rewards but cannot access appropriate content without the algorithm's help, and (2) the platform observes users' strategy profiles—a natural assumption given the scale of user data. In other contexts, users might need to learn their rewards (e.g., in a multi-armed bandit setting [16]), and platforms might need to learn users' strategies as well [15, 32]. A subtle distinction exists between these approaches and our framework. Even in our setting, the platform learns about users during each session, but it makes optimal use of each interaction. Remarkably, we show that these optimal posterior updates admit tractable forms.

Our study also informs the design of interactions with platforms/algorithms by analyzing equilibrium strategies. For example, we quantify how giving users the option to expend extra effort—such as completing a challenging task—can ease the burden of alignment. Such design considerations appear in mechanism design in economics [18] and in human–computer interaction in computer science [28, 29]. For example, breaks during interactions can also promote and sustain long-term engagement [33].

Finally, our work differs from research on preference dynamics [34, 35], where user preferences evolve under the influence of the algorithm. In our setting, users maintain fixed preferences but they strategically adjust some of their actions in response to the algorithm.

## B    Additional statements

**Lemma B.1** (Bellman update). *The algorithm's optimal policy only needs the posterior $[\boldsymbol{\lambda} \mid \widehat{H}; \boldsymbol{f}]$ over user types after observing $\widehat{H}$.*

*Proof.* Overloading the notation, define $Q_{\mathcal{A}}(\boldsymbol{\lambda}, s; \boldsymbol{f}) := \max_{\pi} Q_{\mathcal{A}}(\boldsymbol{\lambda}, s; \boldsymbol{f}, \pi)$. We can expand $Q_{\mathcal{A}}(\boldsymbol{\lambda}, s; \boldsymbol{f})$ as

$$
Q_{\mathcal{A}}(\boldsymbol{\lambda}, s; \boldsymbol{f}) = \mathbb{E}_{\theta \sim \boldsymbol{\lambda}}\Big[\mathbb{E}_{y \sim (f_\theta | s)}\big[y + \gamma_{\mathcal{A}} \max_{s'} \max_{\pi} Q_{\mathcal{A}}\big([\boldsymbol{\lambda} \mid (s, \mathbb{1}\{y > 0\}); \boldsymbol{f}], s'; \boldsymbol{f}, \pi\big)\big]\Big]
$$

$$
= \mathbb{E}_{\theta \sim \boldsymbol{\lambda}}\Big[\frac{f_\theta(s)}{\alpha_\theta(s)} + \gamma_{\mathcal{A}} \mathbb{E}_{\hat{y} \sim \mathrm{Ber}(f_\theta(s))}\big[\max_{s'} Q_{\mathcal{A}}\big([\boldsymbol{\lambda} \mid (s, \hat{y}); \boldsymbol{f}], s'; \boldsymbol{f}\big)\big]\Big].
$$

This shows that the algorithm does not require the entire history for optimal decision making and only needs to update its posterior over user types. $\qquad\square$

**Theorem B.2** (Equilibrium under random entry). *Let $m_\theta^{\mathrm{RE}}$ be the margin of the algorithm's classifier from the perspective of user type $\theta$ when all other user types follow the equilibrium strategy under random entry. When $m_\theta^{\mathrm{RE}} > \lambda_\theta / \alpha_\theta(b)$ and $\gamma_{\mathcal{H}} \neq r_\theta(a)/r_\theta(b)$ for a user of type $\theta \in \Theta_1$, the user's best strategy is*

$$
f_\theta^{\mathrm{RE}}(b) = 1, \quad f_\theta^{\mathrm{RE}}(a) = \mathbb{1}\Big\{\gamma_{\mathcal{H}} < \frac{r_\theta(a)}{r_\theta(b)}\Big\}.
$$

See proof on page 29.

## C    Missing proofs

**Theorem 4.1** (Algorithm's best response). *Given that the algorithm has a posterior $\boldsymbol{\lambda}$ over $\Theta$, it will best respond by recommending item $a$ if and only if $\sum_{\theta \in \Theta} h_\theta \lambda_\theta \geq 0$, where*

$$
h_\theta = \frac{1 - \gamma_{\mathcal{A}}}{(1 - \gamma_{\mathcal{A}} f_\theta(a))(1 - \gamma_{\mathcal{A}} f_\theta(b))}\Big[\frac{f_\theta(a)}{\alpha_\theta(a)} - \frac{f_\theta(b)}{\alpha_\theta(b)}\Big]. \tag{7}
$$

*Proof of Theorem 4.1.* For notational simplicity, we omit $\boldsymbol{f}$ from the notation for $Q_{\mathcal{A}}$. To streamline the proof presentation, we use the following conventions: we use $-s$ to refer to the alternative content to $s$. The inner product of two vectors $\boldsymbol{u}$ and $\boldsymbol{v}$ is denoted by $\langle \boldsymbol{u}, \boldsymbol{v} \rangle$, while the element-wise product is denoted by $\boldsymbol{u} \circ \boldsymbol{v} := (u_\theta v_\theta)_\theta$. When one side of this product is a distribution, the normalized product is defined as $\boldsymbol{u}\hat{\circ}\boldsymbol{v} = \frac{\boldsymbol{u} \circ \boldsymbol{v}}{\langle \boldsymbol{u}, \boldsymbol{v} \rangle}$. Division, such as $\boldsymbol{u}/\boldsymbol{v}$, denotes element-wise division.

Using this notation, we can rewrite the algorithm's Bellman update from Eq. (1) as follows. First, the posterior $[\boldsymbol{\lambda} \mid (s, \hat{y})]$ simplifies to the following form:

$$
[\boldsymbol{\lambda} \mid (s, \hat{y})] = \begin{cases} \boldsymbol{\lambda}\hat{\circ}\boldsymbol{f}(s), & \hat{y} = 1, \\ \boldsymbol{\lambda}\hat{\circ}(1 - \boldsymbol{f}(s)), & \hat{y} = 0. \end{cases}
$$

We can also express the expected immediate reward $\mathbb{E}_\theta[f_\theta(s)/\alpha_\theta(s)]$ as $\langle \boldsymbol{\lambda}, \boldsymbol{f}(s)/\boldsymbol{\alpha}(s) \rangle$. Using this, the Bellman update for $Q_{\mathcal{A}}$ becomes

$$
\begin{aligned}
Q_{\mathcal{A}}(\boldsymbol{\lambda}, s) = &\langle \boldsymbol{\lambda}, \boldsymbol{f}(s)/\boldsymbol{\alpha}(s) \rangle \\
&+ \gamma_{\mathcal{A}} \langle \boldsymbol{\lambda}, \boldsymbol{f}(s) \rangle \max_{s'} Q_{\mathcal{A}}\big(\boldsymbol{\lambda}\hat{\circ}\boldsymbol{f}(s), s'\big) \\
&+ \gamma_{\mathcal{A}} \langle \boldsymbol{\lambda}, (1 - \boldsymbol{f}(s)) \rangle \max_{s'} Q_{\mathcal{A}}\big(\boldsymbol{\lambda}\hat{\circ}(1 - \boldsymbol{f}(s)), s'\big).
\end{aligned} \tag{14}
$$

Note that $\boldsymbol{\lambda}\hat{\circ}\boldsymbol{f}(s)$ or $\boldsymbol{\lambda}\hat{\circ}(1 - \boldsymbol{f}(s))$ may be undefined if all users choose to either fully engage or fully disengage. However, since the second and third terms are also multiplied by $\langle \boldsymbol{\lambda}, \boldsymbol{f}(s) \rangle$ and $\langle \boldsymbol{\lambda}, (1 - \boldsymbol{f}(s)) \rangle$ respectively, this issue can be neglected. We prove that the following Q-function solves Eq. (14):

$$
Q_{\mathcal{A}}(\boldsymbol{\lambda}, s) = \langle \boldsymbol{\lambda}, \boldsymbol{q}(s) \rangle + \gamma_{\mathcal{A}} \max\Big\{ \langle \boldsymbol{\lambda}, (\boldsymbol{q}(-s) - \boldsymbol{q}(s)) \circ \boldsymbol{f}(s) \rangle, 0 \Big\}, \tag{15}
$$

where

$$\boldsymbol{q}(s) := \frac{\boldsymbol{f}(s)}{1 - \gamma_{\mathcal{A}}\boldsymbol{f}(s)} \circ \frac{1}{\boldsymbol{\alpha}(s)} + \gamma_{\mathcal{A}} \frac{\boldsymbol{f}(-s) \circ (1 - \boldsymbol{f}(s))}{(1 - \gamma_{\mathcal{A}}\boldsymbol{f}(s)) \circ (1 - \gamma_{\mathcal{A}}\boldsymbol{f}(-s))} \circ \frac{1}{\boldsymbol{\alpha}(-s)}. \tag{16}$$

One can verify that $\boldsymbol{h}$ in Eq. (7) can be expressed as $\boldsymbol{q}(a) - \boldsymbol{q}(b)$. Before proving Eq. (15), we first show that it implies $\boldsymbol{h} := \boldsymbol{q}(a) - \boldsymbol{q}(b)$ serves as the linear classifier that determines the algorithm's policy:

**Lemma C.1.** $Q_{\mathcal{A}}(\boldsymbol{\lambda}, s) \geq Q_{\mathcal{A}}(\boldsymbol{\lambda}, -s) \iff \langle \boldsymbol{\lambda}, \boldsymbol{q}(s) - \boldsymbol{q}(-s) \rangle \geq 0$.

See proof on page 30. The proof of this lemma relies on the following lemma, which we will use again later on.

**Lemma C.2.** $\langle \boldsymbol{\lambda} \circ \boldsymbol{f}(s), \boldsymbol{q}(s) - \boldsymbol{q}(-s) \rangle \geq \langle \boldsymbol{\lambda}, \boldsymbol{q}(s) - \boldsymbol{q}(-s) \rangle$.

See proof on page 30. This lemma also yields the following result that is useful in simplifying the second term of Eq. (14):

**Lemma C.3.** $\max_{s'} Q_{\mathcal{A}}(\boldsymbol{\lambda}, s') = \max_{s'} \langle \boldsymbol{\lambda}, \boldsymbol{q}(s') \rangle$.

See proof on page 31. Together Lemmas C.1 to C.3 give the following result that is useful in simplifying the third term of Eq. (14):

**Lemma C.4.** If $\langle \boldsymbol{\lambda}, 1 - \boldsymbol{f}(s) \rangle > 0$, we have $\max_{s'} Q_{\mathcal{A}}(\boldsymbol{\lambda}\hat{\circ}(1 - \boldsymbol{f}(s)), s') = \langle \boldsymbol{\lambda}\hat{\circ}(1 - \boldsymbol{f}(s)), \boldsymbol{q}(-s) \rangle$.

See proof on page 31.

Using Lemmas C.3 and C.4 we can write the right-hand side of Eq. (14) as

$$\langle \boldsymbol{\lambda}, \boldsymbol{f}(s)/\boldsymbol{\alpha}(s) \rangle + \gamma_{\mathcal{A}} \langle \boldsymbol{\lambda}, \boldsymbol{f}(s) \rangle \max_{s'} \langle \boldsymbol{\lambda}\hat{\circ}\boldsymbol{f}(s), \boldsymbol{q}(s') \rangle + \gamma_{\mathcal{A}} \langle \boldsymbol{\lambda}, 1 - \boldsymbol{f}(s) \rangle \langle \boldsymbol{\lambda}\hat{\circ}(1 - \boldsymbol{f}(s)), \boldsymbol{q}(-s) \rangle$$

$$= \langle \boldsymbol{\lambda}, \boldsymbol{f}(s)/\boldsymbol{\alpha}(s) \rangle + \gamma_{\mathcal{A}} \max_{s'} \langle \boldsymbol{\lambda} \circ \boldsymbol{f}(s), \boldsymbol{q}(s') \rangle + \gamma_{\mathcal{A}} \langle \boldsymbol{\lambda} \circ (1 - \boldsymbol{f}(s)), \boldsymbol{q}(-s) \rangle$$

$$= \langle \boldsymbol{\lambda}, \boldsymbol{f}(s)/\boldsymbol{\alpha}(s) \rangle + \gamma_{\mathcal{A}} \max \left\{ \langle \boldsymbol{\lambda} \circ \boldsymbol{f}(s), \boldsymbol{q}(-s) - \boldsymbol{q}(s) \rangle, 0 \right\} + \gamma_{\mathcal{A}} \langle \boldsymbol{\lambda}, \boldsymbol{q}(s) \circ \boldsymbol{f}(s) + \boldsymbol{q}(-s) \circ (1 - \boldsymbol{f}(s)) \rangle. \tag{17}$$

Using $(1 - \boldsymbol{f}(s))(1 - \boldsymbol{f}(-s)) = 0$ from Assumption 1, we can further simplify the first and third (last) terms by

$$\frac{\boldsymbol{f}(s)}{\boldsymbol{\alpha}(s)} + \gamma_{\mathcal{A}}\boldsymbol{q}(s) \circ \boldsymbol{f}(s) + \gamma_{\mathcal{A}}\boldsymbol{q}(-s) \circ (1 - \boldsymbol{f}(s))$$

$$= \frac{1}{\boldsymbol{\alpha}(s)} \circ \left[ \boldsymbol{f}(s) + \gamma_{\mathcal{A}} \frac{\boldsymbol{f}^2(s)}{1 - \gamma_{\mathcal{A}}\boldsymbol{f}(s)} \right]$$

$$+ \gamma_{\mathcal{A}} \frac{1}{\boldsymbol{\alpha}(-s)} \circ \left[ \gamma_{\mathcal{A}} \frac{\boldsymbol{f}(s) \circ \boldsymbol{f}(-s) \circ (1 - \boldsymbol{f}(s))}{(1 - \gamma_{\mathcal{A}}\boldsymbol{f}(s)) \circ (1 - \gamma_{\mathcal{A}}\boldsymbol{f}(-s))} + \frac{\boldsymbol{f}(-s) \circ (1 - \boldsymbol{f}(s))}{1 - \gamma_{\mathcal{A}}\boldsymbol{f}(-s)} \right]$$

$$= \frac{1}{\boldsymbol{\alpha}(s)} \circ \frac{\boldsymbol{f}(s)}{1 - \gamma_{\mathcal{A}}\boldsymbol{f}(s)} + \gamma_{\mathcal{A}} \frac{1}{\boldsymbol{\alpha}(-s)} \circ \frac{\boldsymbol{f}(-s) \circ (1 - \boldsymbol{f}(s))}{(1 - \gamma_{\mathcal{A}}\boldsymbol{f}(s)) \circ (1 - \gamma_{\mathcal{A}}\boldsymbol{f}(-s))} = \boldsymbol{q}(s).$$

Plugging this into Eq. (17) gives $Q_{\mathcal{A}}(\boldsymbol{\lambda}, s)$ as defined in Eq. (15). Therefore, the proposed $Q_{\mathcal{A}}$ solves the Bellman update of Eq. (14). This completes the proof. $\square$

**Theorem 5.1** (Equilibrium under algorithmic entry). *Let $m_{\theta}^{\mathrm{AE}}$ be the margin of the algorithm's classifier from the perspective of user type $\theta$ when all other user types follow the equilibrium strategy under algorithmic entry. Define the* steerable sets *for type 1 and 2 users as follows:*

$$\theta \in \Theta_1: \; F_{\theta} := \left\{ x \in \left[0, \frac{\alpha_{\theta}(a)}{\alpha_{\theta}(b)}\right) \mid \frac{\lambda_{\theta}}{\alpha_{\theta}(b)} - m_{\theta}^{\mathrm{AE}} - x \left(\frac{\lambda_{\theta}}{\alpha_{\theta}(a)} - \gamma_{\mathcal{A}} m_{\theta}^{\mathrm{AE}}\right) \geq 0 \right\}$$

$$\theta \in \Theta_2: \; F_{\theta} := \left\{ x \in [0, 1] \mid \frac{\lambda_{\theta}}{\alpha_{\theta}(a)} + m_{\theta}^{\mathrm{AE}} - x \left(\frac{\lambda_{\theta}}{\alpha_{\theta}(b)} + \gamma_{\mathcal{A}} m_{\theta}^{\mathrm{AE}}\right) \geq 0 \right\}.$$

*Let $s^*_\theta$ and $(-s^*_\theta)$ be the high and low reward contents for type $\theta$. The user's strategy at equilibrium is*

$$f^{\mathrm{AE}}_\theta(s^*_\theta) = 1\,, \quad f^{\mathrm{AE}}_\theta(-s^*_\theta) = \begin{cases} \text{any value in } F_\theta\,, & F_\theta \neq \emptyset\,, \\ 0\,, & F_\theta = \emptyset\,, \gamma_{\mathcal{H}} > \frac{r_\theta(-s^*_\theta)}{r_\theta(s^*_\theta)}\,, \\ 1\,, & F_\theta = \emptyset\,, \gamma_{\mathcal{H}} < \frac{r_\theta(-s^*_\theta)}{r_\theta(s^*_\theta)}\,, \\ \text{any value in } [0,1]\,, & F_\theta = \emptyset\,, \gamma_{\mathcal{H}} = \frac{r_\theta(-s^*_\theta)}{r_\theta(s^*_\theta)}\,. \end{cases}$$

*Proof of Theorem 5.1.* We use a similar notation as in the proof of Theorem 4.1. For improved readability, we drop $\boldsymbol{f}$ from $Q_\theta(\boldsymbol{\lambda}, s; \boldsymbol{f}, \boldsymbol{f})$ and $Q_{\mathcal{A}}(\boldsymbol{\lambda}, s; \boldsymbol{f})$. With this notation, the Bellman update for user type $\theta$ in Eq. (2) can be written as

$$Q_\theta(\boldsymbol{\lambda}, s) = f_\theta(s)\, r_\theta(s)$$
$$+ \gamma_{\mathcal{H}} f_\theta(s)\, Q_\theta\Big(\boldsymbol{\lambda}\hat{\circ}\boldsymbol{f}(s), \arg\max_{s'} Q_{\mathcal{A}}\big(\boldsymbol{\lambda}\hat{\circ}\boldsymbol{f}(s), s'\big)\Big)$$
$$+ \gamma_{\mathcal{H}}(1 - f_\theta(s))\, Q_\theta\Big(\boldsymbol{\lambda}\hat{\circ}(1 - \boldsymbol{f}(s)), \arg\max_{s'} Q_{\mathcal{A}}\big(\boldsymbol{\lambda}\hat{\circ}(1 - \boldsymbol{f}(s)), s'\big)\Big)\,.$$

Since the user's entry and subsequent interactions occur under the algorithm's best response, we only need to solve the above for $s \in \arg\max_{s'} Q_{\mathcal{A}}(\boldsymbol{\lambda}, s')$. When $Q_{\mathcal{A}}(\boldsymbol{\lambda}, s) \geq Q_{\mathcal{A}}(\boldsymbol{\lambda}, -s)$, Lemmas C.1 and C.2 imply

$$s \in \arg\max_{s'} Q_{\mathcal{A}}\big(\boldsymbol{\lambda}\hat{\circ}\boldsymbol{f}(s), s'\big)\,,$$
$$(-s) \in \arg\max_{s'} Q_{\mathcal{A}}\big(\boldsymbol{\lambda}\hat{\circ}(1 - \boldsymbol{f}(s)), s'\big)\,.$$

Plugging these into the Bellman update, we obtain

$$Q_\theta(\boldsymbol{\lambda}, s) = f_\theta(s)\, r_\theta(s) + \gamma_{\mathcal{H}} f_\theta(s)\, Q_\theta\big(\boldsymbol{\lambda}\hat{\circ}\boldsymbol{f}(s), s\big) + \gamma_{\mathcal{H}}(1 - f_\theta(s))\, Q_\theta\big(\boldsymbol{\lambda}\hat{\circ}(1 - \boldsymbol{f}(s)), -s\big)\,.$$

Note that this equation has no dependence on $\boldsymbol{\lambda}$, so, we can drop it from the notation. Using Assumption 1, we can write the above update separately for $s = s^*_\theta$ and $s = (-s^*_\theta)$:

$$Q_\theta(s^*_\theta) = r_\theta(s^*_\theta) + \gamma_{\mathcal{H}}\, Q_\theta(s^*_\theta)\,,$$
$$Q_\theta(-s^*_\theta) = f_\theta(-s^*_\theta)\, r_\theta(-s^*_\theta) + \gamma_{\mathcal{H}} f_\theta(-s^*_\theta)\, Q_\theta(-s^*_\theta) + \gamma_{\mathcal{H}}(1 - f_\theta(-s^*_\theta))\, Q_\theta(s^*_\theta)\,.$$

Solving these equations, we obtain

$$Q_\theta(s^*_\theta) = \frac{r_\theta(s^*_\theta)}{1 - \gamma_{\mathcal{H}}}\,,$$
$$Q_\theta(-s^*_\theta) = \frac{f_\theta(-s^*_\theta)\, r_\theta(-s^*_\theta)}{1 - \gamma_{\mathcal{H}} f_\theta(-s^*_\theta)} + \frac{\gamma_{\mathcal{H}}(1 - f_\theta(-s^*_\theta))\, r_\theta(s^*_\theta)}{(1 - \gamma_{\mathcal{H}} f_\theta(-s^*_\theta))\,(1 - \gamma_{\mathcal{H}})}$$
$$= \frac{r_\theta(s^*_\theta)}{1 - \gamma_{\mathcal{H}}} - \frac{r_\theta(s^*_\theta) - f_\theta(-s^*_\theta)\, r_\theta(-s^*_\theta)}{1 - \gamma_{\mathcal{H}} f_\theta(-s^*_\theta)}\,.$$

Starting from a prior $\boldsymbol{\lambda}$ over user types, the user's value is

$$V_\theta(\boldsymbol{\lambda}) := Q_\theta\big(\arg\max_s Q_{\mathcal{A}}(\boldsymbol{\lambda}, s)\big)\,.$$

Given $V_\theta$, we now explore the user's best strategy that maximizes $V_\theta$, leading to the equilibrium notion defined in Eq. (4). Note that $Q_\theta(s^*_\theta) \geq Q_\theta(-s^*_\theta)$. Therefore, the optimal strategy is to select $f_\theta(-s^*_\theta)$ such that $s^*_\theta \in \arg\max_s Q_{\mathcal{A}}(\boldsymbol{\lambda}, s)$. In the equilibrium, Theorem 4.1 implies that this is only possible when

$$h_\theta(s^*_\theta)\lambda_\theta = \frac{\lambda_\theta}{1 - \gamma_{\mathcal{A}} f_\theta(-s^*_\theta)}\Big[\frac{1}{\alpha_\theta(s^*_\theta)} - \frac{f_\theta(-s^*_\theta)}{\alpha_\theta(-s^*_\theta)}\Big] \geq -\big\langle \boldsymbol{\lambda}_{-\theta}, \boldsymbol{h}^{\mathrm{AE}}_{-\theta}(s^*_\theta)\big\rangle\,. \tag{18}$$

Here, we generalized $\boldsymbol{h}$ in Eq. (7) by defining $\boldsymbol{h}(s) = \mathbb{1}\{s = a\} \cdot \boldsymbol{h} - \mathbb{1}\{s = b\} \cdot \boldsymbol{h}$. Eq. (18) is a linear constraint over $f_\theta(-s^*_\theta)$:

$$\frac{\lambda_\theta}{\alpha_\theta(s^*_\theta)} + \big\langle \boldsymbol{\lambda}_{-\theta}, \boldsymbol{h}^{\mathrm{AE}}_{-\theta}(s^*_\theta)\big\rangle - f_\theta(-s^*_\theta)\Big(\frac{\lambda_\theta}{\alpha_\theta(-s^*_\theta)} + \gamma_{\mathcal{A}}\big\langle \boldsymbol{\lambda}_{-\theta}, \boldsymbol{h}^{\mathrm{AE}}_{-\theta}(s^*_\theta)\big\rangle\Big) \geq 0\,.$$

Any $f_\theta(-s_\theta^*)$ that meets this condition is the user's best response. If the condition does not hold, then $V_\theta(\boldsymbol{\lambda}) = Q_\theta(-s_\theta^*)$. In this case, the user's best response is

$$f_\theta^{\mathrm{AE}}(-s_\theta^*) = \begin{cases} 0\,, & \gamma_{\mathcal{H}} > \frac{r_\theta(-s_\theta^*)}{r_\theta(s_\theta^*)}\,, \\ 1\,, & \gamma_{\mathcal{H}} < \frac{r_\theta(-s_\theta^*)}{r_\theta(s_\theta^*)}\,, \\ [0,1] & \text{o.w.} \end{cases}$$

Using specific values for type 1 and type 2 users above will complete the proof. $\qquad\square$

**Corollary 5.2.** *The steerable set for user type $\theta \in \Theta_1$ is nonempty if and only if $\lambda_\theta \geq \alpha_\theta(b)\, m_\theta^{\mathrm{AE}}$. Moreover, when nonempty, $F_\theta = [0, c)$ for some $c$.*

*Proof of Corollary 5.2.* Suppose $\lambda_\theta < \gamma_{\mathcal{A}}\, \alpha_\theta(a)\, m_\theta^{\mathrm{AE}}$. In this case,

$$\frac{\lambda_\theta}{\alpha_\theta(b)} - m_\theta^{\mathrm{AE}} - x\left(\frac{\lambda_\theta}{\alpha_\theta(a)} - \gamma_{\mathcal{A}}\, m_\theta^{\mathrm{AE}}\right) < -\lambda_\theta\left(\frac{1}{\alpha_\theta(a)} - \frac{1}{\alpha_\theta(b)}\right) - m_\theta^{\mathrm{AE}}\,(1 - \gamma_{\mathcal{A}}) \leq 0\,,$$

which means $F_\theta = \emptyset$. Therefore,

$$F_\theta \neq \emptyset \implies \lambda_\theta \geq \gamma_{\mathcal{A}}\, \alpha_\theta(a)\, m_\theta^{\mathrm{AE}}\,.$$

Now suppose $x \in F_\theta$. Then the above implies any $x' \leq x$ is also in $F_\theta$. Therefore, when $F_\theta \neq \emptyset$, it covers $[0, c)$ for some $c$. Particularly, when $F_\theta \neq \emptyset$, we should have $0 \in F_\theta$ which implies

$$F_\theta \neq \emptyset \implies \lambda_\theta \geq \alpha_\theta(b)\, m_\theta^{\mathrm{AE}}\,.$$

Now suppose $\lambda_\theta \geq \alpha_\theta(b)\, m_\theta^{\mathrm{AE}}$. Then $0 \in F_\theta$, so

$$\lambda_\theta \geq \alpha_\theta(b)\, m_\theta^{\mathrm{AE}} \implies F_\theta \neq \emptyset\,.$$

Thus, we can conclude

$$\lambda_\theta \geq \alpha_\theta(b)\, m_\theta^{\mathrm{AE}} \iff F_\theta \neq \emptyset\,.$$

$\qquad\square$

**Corollary 5.4.** *A user of type $\theta \in \Theta_1$ has constant regret in equilibrium if and only if Eq. (9) holds.*

*Proof of Corollary 5.4.* First, there always exists an equilibrium where the conditions of Corollary 5.3 and Theorem B.2 are satisfied. This occurs, for example, when type 1 users with inconsistent actions and interests form a small part of the population. In such cases, the optimal strategy for a type 1 user is either $f_\theta(a) = 0$ or $f_\theta(a) = 1$. When the user fully engages, the algorithm cannot distinguish her from users with aligned interests, leading it to continue recommending type $b$ content. In contrast, when the user fully disengages, the algorithm will recommend type $b$ content at most once. $\qquad\square$

**Theorem 6.1** (Algorithm's best response with signaling)**.** *Given that the algorithm has a posterior $\boldsymbol{\lambda}$ over $\Theta$, it will best respond by recommending item $a$ if and only if $\sum_{\theta \in \Theta} h_\theta \lambda_\theta \geq 0$, where*

$$h_\theta = \frac{1 - \gamma_{\mathcal{A}}}{\left(1 - \gamma_{\mathcal{A}} f_\theta(a)(1 - u_\theta(a))\right)\left(1 - \gamma_{\mathcal{A}} f_\theta(b)(1 - u_\theta(b))\right)}\left[\frac{f_\theta(a)}{\alpha_\theta(a)} - \frac{f_\theta(b)}{\alpha_\theta(b)}\right]\,. \tag{13}$$

*Proof of Theorem 6.1.* We follow a similar notation as in the proof of Theorem 4.1. Using this notation, we can rewrite the algorithm's Bellman update from Eq. (10) as follows. First, the posterior $[\boldsymbol{\lambda} \mid (s, \hat{y}, \hat{u})]$ simplifies to the following form:

$$[\boldsymbol{\lambda} \mid (s, \hat{y}, \hat{u})] = \begin{cases} \boldsymbol{\lambda} \hat{\circ}[\boldsymbol{u}(s) \circ \boldsymbol{f}(s)]\,, & \hat{u} = 1, \hat{y} = 1\,, \\ \boldsymbol{\lambda} \hat{\circ}[\boldsymbol{u}(s) \circ (1 - \boldsymbol{f}(s))]\,, & \hat{u} = 1, \hat{y} = 0\,, \\ \boldsymbol{\lambda} \hat{\circ}[(1 - \boldsymbol{u}(s)) \circ \boldsymbol{f}(s)]\,, & \hat{u} = 0, \hat{y} = 1\,, \\ \boldsymbol{\lambda} \hat{\circ}[(1 - \boldsymbol{u}(s)) \circ (1 - \boldsymbol{f}(s))]\,, & \hat{u} = 0, \hat{y} = 0\,. \end{cases}$$

We can also express the expected immediate reward $\mathbb{E}_\theta[f_\theta(s)/\alpha_\theta(s)]$ as $\langle \boldsymbol{\lambda}, \boldsymbol{f}(s)/\boldsymbol{\alpha}(s)\rangle$. Using this, the Bellman update for $Q_\mathcal{A}$ becomes

$$
\begin{aligned}
Q_\mathcal{A}(\boldsymbol{\lambda}, s) = {} & \langle \boldsymbol{\lambda}, \boldsymbol{f}(s)/\boldsymbol{\alpha}(s)\rangle \\
& + \gamma_\mathcal{A}\, \langle \boldsymbol{\lambda}, \boldsymbol{u} \circ \boldsymbol{f}(s)\rangle \max_{s'} Q_\mathcal{A}\big(\boldsymbol{\lambda}\hat{\circ}[\boldsymbol{u}(s) \circ \boldsymbol{f}(s)], s'\big) \\
& + \gamma_\mathcal{A}\, \langle \boldsymbol{\lambda}, \boldsymbol{u} \circ (1 - \boldsymbol{f}(s))\rangle \max_{s'} Q_\mathcal{A}\big(\boldsymbol{\lambda}\hat{\circ}[\boldsymbol{u}(s) \circ (1 - \boldsymbol{f}(s))], s'\big) \\
& + \gamma_\mathcal{A}\, \langle \boldsymbol{\lambda}, (1 - \boldsymbol{u}) \circ \boldsymbol{f}(s)\rangle \max_{s'} Q_\mathcal{A}\big(\boldsymbol{\lambda}\hat{\circ}[(1 - \boldsymbol{u}(s)) \circ \boldsymbol{f}(s)], s'\big) \\
& + \gamma_\mathcal{A}\, \langle \boldsymbol{\lambda}, (1 - \boldsymbol{u}) \circ (1 - \boldsymbol{f}(s))\rangle \max_{s'} Q_\mathcal{A}\big(\boldsymbol{\lambda}\hat{\circ}[(1 - \boldsymbol{u}(s)) \circ (1 - \boldsymbol{f}(s))], s'\big) .
\end{aligned}
\tag{19}
$$

We prove that the following Q-function solves the above:

$$
Q_\mathcal{A}(\boldsymbol{\lambda}, s) = \langle \boldsymbol{\lambda}, \boldsymbol{q}(s)\rangle + \gamma_\mathcal{A}\, \max\Big\{ \langle \boldsymbol{\lambda}, (\boldsymbol{q}(-s) - \boldsymbol{q}(s)) \circ (1 - \boldsymbol{u}(s)) \circ \boldsymbol{f}(s)\rangle , 0 \Big\},
\tag{20}
$$

where

$$
\boldsymbol{q}(s) := \frac{\boldsymbol{f}(s)}{1 - \gamma_\mathcal{A}(1 - \boldsymbol{u}(s)) \circ \boldsymbol{f}(s)} \circ \frac{1}{\boldsymbol{\alpha}(s)} + \gamma_\mathcal{A} \frac{\boldsymbol{f}(-s) \circ \big(1 - (1 - \boldsymbol{u}(s)) \circ \boldsymbol{f}(s)\big)}{(1 - \gamma_\mathcal{A}(1 - \boldsymbol{u}(s)) \circ \boldsymbol{f}(s)) \circ (1 - \gamma_\mathcal{A}\boldsymbol{f}(-s))} \circ \frac{1}{\boldsymbol{\alpha}(-s)} .
\tag{21}
$$

Using $\boldsymbol{u}(s) \circ \boldsymbol{u}(-s) = 0$ and $\boldsymbol{u}(s) \circ (1 - \boldsymbol{f}(-s)) = 0$ implied by Assumption 2, one can verify that $\boldsymbol{h}$ in Eq. (13) can be expressed as $\boldsymbol{q}(a) - \boldsymbol{q}(b)$. Before proving Eq. (20), we first show that it implies $\boldsymbol{h} := \boldsymbol{q}(a) - \boldsymbol{q}(b)$ serves as the linear classifier that determines the algorithm's policy:

**Lemma C.5.** $Q_\mathcal{A}(\boldsymbol{\lambda}, s) \geq Q_\mathcal{A}(\boldsymbol{\lambda}, -s) \iff \langle \boldsymbol{\lambda}, \boldsymbol{q}(s) - \boldsymbol{q}(-s)\rangle \geq 0$.

See proof on page 31. The proof of this lemma relies on Lemma C.2 and the following lemma:

**Lemma C.6.** $\langle \boldsymbol{\lambda} \circ (1 - \boldsymbol{u}(s)), \boldsymbol{q}(s) - \boldsymbol{q}(-s)\rangle \geq \langle \boldsymbol{\lambda}, \boldsymbol{q}(s) - \boldsymbol{q}(-s)\rangle$ .

See proof on page 31. This lemma also yields the following result that is useful in simplifying the fourth term of Eq. (19):

**Lemma C.7.** $\max_{s'} Q_\mathcal{A}(\boldsymbol{\lambda}, s') = \max_{s'} \langle \boldsymbol{\lambda}, \boldsymbol{q}(s')\rangle$ .

See proof on page 32. Together Lemmas C.1 to C.3 give the following result that is useful in simplifying the first and second term of Eq. (14):

**Lemma C.8.** *If* $\langle \boldsymbol{\lambda}, \boldsymbol{u}(s) \circ \boldsymbol{g}(s)\rangle > 0$ *for some* $\boldsymbol{g}$, *we have*

$$
\max_{s'} Q_\mathcal{A}\big(\boldsymbol{\lambda}\hat{\circ}[\boldsymbol{u}(s) \circ \boldsymbol{g}(s)], s'\big) = \langle \boldsymbol{\lambda}\hat{\circ}[\boldsymbol{u}(s) \circ \boldsymbol{g}(s)], \boldsymbol{q}(-s)\rangle .
$$

See proof on page 32.

Using Lemma C.8 for the second and third term, Lemma C.7 for the fourth term, and Lemma C.4 for the fifth term in the right-hand side of Eq. (19), we can simplify the Bellman update as

$$
\begin{aligned}
& \langle \boldsymbol{\lambda}, \boldsymbol{f}(s)/\boldsymbol{\alpha}(s)\rangle + \gamma_\mathcal{A}\, \langle \boldsymbol{\lambda}, (1 - \boldsymbol{u}(s)) \circ \boldsymbol{f}(s)\rangle \max_{s'} \langle \boldsymbol{\lambda}\hat{\circ}[(1 - \boldsymbol{u}(s)) \circ \boldsymbol{f}(s)], \boldsymbol{q}(s')\rangle \\
& + \gamma_\mathcal{A}\, \langle \boldsymbol{\lambda} \circ (1 - \boldsymbol{f}(s) + \boldsymbol{u}(s) \circ \boldsymbol{f}(s)), \boldsymbol{q}(-s)\rangle \\
={} & \langle \boldsymbol{\lambda}, \boldsymbol{f}(s)/\boldsymbol{\alpha}(s)\rangle + \gamma_\mathcal{A}\, \max\Big\{ \langle \boldsymbol{\lambda} \circ (1 - \boldsymbol{u}(s)) \circ \boldsymbol{f}(s), \boldsymbol{q}(-s) - \boldsymbol{q}(s)\rangle , 0 \Big\} \\
& + \gamma_\mathcal{A}\, \langle \boldsymbol{\lambda}, \boldsymbol{q}(s) \circ (1 - \boldsymbol{u}(s)) \circ \boldsymbol{f}(s) + \boldsymbol{q}(-s) \circ (1 - \boldsymbol{f}(s) + \boldsymbol{u}(s) \circ \boldsymbol{f}(s))\rangle .
\end{aligned}
\tag{22}
$$

Using $(1 - \boldsymbol{f}(s))(1 - \boldsymbol{f}(-s)) = 0$ from Assumption 1, we can further simplify the first and third (last) terms by

$$
\begin{aligned}
& \frac{\boldsymbol{f}(s)}{\boldsymbol{\alpha}(s)} + \gamma_\mathcal{A}\boldsymbol{q}(s) \circ (1 - \boldsymbol{u}(s)) \circ \boldsymbol{f}(s) + \gamma_\mathcal{A}\boldsymbol{q}(-s) \circ (1 - \boldsymbol{f}(s) + \boldsymbol{u}(s) \circ \boldsymbol{f}(s)) \\
={} & \frac{1}{\boldsymbol{\alpha}(s)} \circ \left[ \boldsymbol{f}(s) + \gamma_\mathcal{A} \frac{(1 - \boldsymbol{u}(s)) \circ \boldsymbol{f}^2(s)}{1 - \gamma_\mathcal{A}(1 - \boldsymbol{u}(s)) \circ \boldsymbol{f}(s)} \right] \\
& + \gamma_\mathcal{A} \frac{1}{\boldsymbol{\alpha}(-s)} \circ \left[ \gamma_\mathcal{A} \frac{(1 - \boldsymbol{u}(s)) \circ \boldsymbol{f}(s) \circ \boldsymbol{f}(-s) \circ \big(1 - (1 - \boldsymbol{u}(s)) \circ \boldsymbol{f}(s)\big)}{(1 - \gamma_\mathcal{A}(1 - \boldsymbol{u}(s)) \circ \boldsymbol{f}(s)) \circ (1 - \gamma_\mathcal{A}\boldsymbol{f}(-s))} + \frac{\boldsymbol{f}(-s) \circ \big(1 - (1 - \boldsymbol{u}(s)) \circ \boldsymbol{f}(s)\big)}{1 - \gamma_\mathcal{A}(1 - \boldsymbol{u}(-s)) \circ \boldsymbol{f}(-s)} \right] .
\end{aligned}
\tag{23}
$$

The following property implied by Assumptions 1 and 2 are useful to further simplify the above:

$$\boldsymbol{u}(-s) > 0 \implies 1 - (1 - \boldsymbol{u}(s)) \circ \boldsymbol{f}(s) = 0$$

Using this in Eq. (23), we obtain

$$\frac{1}{\boldsymbol{\alpha}(s)} \circ \frac{\boldsymbol{f}(s)}{1 - \gamma_\mathcal{A}(1 - \boldsymbol{u}(s)) \circ \boldsymbol{f}(s)}$$

$$+ \gamma_\mathcal{A} \frac{1}{\boldsymbol{\alpha}(-s)} \circ \left[ \gamma_\mathcal{A} \frac{(1 - \boldsymbol{u}(s)) \circ \boldsymbol{f}(s) \circ \boldsymbol{f}(-s) \circ \left(1 - (1 - \boldsymbol{u}(s)) \circ \boldsymbol{f}(s)\right)}{(1 - \gamma_\mathcal{A}(1 - \boldsymbol{u}(s)) \circ \boldsymbol{f}(s)) \circ (1 - \gamma_\mathcal{A} \boldsymbol{f}(-s))} + \frac{\boldsymbol{f}(-s) \circ \left(1 - (1 - \boldsymbol{u}(s)) \circ \boldsymbol{f}(s)\right)}{1 - \gamma_\mathcal{A} \boldsymbol{f}(-s)} \right]$$

$$= \frac{1}{\boldsymbol{\alpha}(s)} \circ \frac{\boldsymbol{f}(s)}{1 - \gamma_\mathcal{A}(1 - \boldsymbol{u}(s)) \circ \boldsymbol{f}(s)}$$

$$+ \gamma_\mathcal{A} \frac{1}{\boldsymbol{\alpha}(-s)} \circ \boldsymbol{f}(-s) \circ \left(1 - (1 - \boldsymbol{u}(s)) \circ \boldsymbol{f}(s)\right) \frac{\gamma_\mathcal{A}(1 - \boldsymbol{u}(s)) \circ \boldsymbol{f}(s) + 1 - \gamma_\mathcal{A}(1 - \boldsymbol{u}(s)) \circ \boldsymbol{f}(s)}{(1 - \gamma_\mathcal{A}(1 - \boldsymbol{u}(s)) \circ \boldsymbol{f}(s)) \circ (1 - \gamma_\mathcal{A} \boldsymbol{f}(-s))}$$

$$= \frac{1}{\boldsymbol{\alpha}(s)} \circ \frac{\boldsymbol{f}(s)}{1 - \gamma_\mathcal{A}(1 - \boldsymbol{u}(s)) \circ \boldsymbol{f}(s)} + \gamma_\mathcal{A} \frac{1}{\boldsymbol{\alpha}(-s)} \circ \frac{\boldsymbol{f}(-s) \circ \left(1 - (1 - \boldsymbol{u}(s)) \circ \boldsymbol{f}(s)\right)}{(1 - \gamma_\mathcal{A}(1 - \boldsymbol{u}(s)) \circ \boldsymbol{f}(s)) \circ (1 - \gamma_\mathcal{A} \boldsymbol{f}(-s))}$$

$$= \boldsymbol{q}(s) \,.$$

Plugging this into Eq. (22) gives $Q_\mathcal{A}(\boldsymbol{\lambda}, s)$ as defined in Eq. (20). Hence, the proposed $Q_\mathcal{A}$ solves the Bellman update of Eq. (19). This completes the proof. $\qquad\square$

**Theorem 6.2** (Equilibrium under algorithmic entry with signaling). *Let $m_\theta^{\mathrm{AE}}$ be the margin of the algorithm's classifier from the perspective of user type $\theta$ when other user types follow the equilibrium strategy under algorithmic entry with signaling. Define the* steerable sets *for type 1 and 2 users as*

$$\theta \in \Theta_1: \ F_\theta := \left\{ (x, y) \in \left[0, \frac{\alpha_\theta(a)}{\alpha_\theta(b)}\right) \times [0, 1] \ \Big| \ \frac{\lambda_\theta}{\alpha_\theta(b)} - m_\theta^{\mathrm{AE}} - x \left( \frac{\lambda_\theta}{\alpha_\theta(a)} - \gamma_\mathcal{A} (1 - y) m_\theta^{\mathrm{AE}} \right) \geq 0 \right\}$$

$$\theta \in \Theta_2: \ F_\theta := \left\{ (x, y) \in [0, 1]^2 \ \Big| \ \frac{\lambda_\theta}{\alpha_\theta(a)} + m_\theta^{\mathrm{AE}} - x \left( \frac{\lambda_\theta}{\alpha_\theta(b)} + \gamma_\mathcal{A} (1 - y) m_\theta^{\mathrm{AE}} \right) \geq 0 \right\}.$$

*Let $s_\theta^*$ and $(-s_\theta^*)$ be the high and low reward contents for type $\theta$. Define the critical $\gamma_\mathcal{H}$ value for type $\theta$ as*

$$\gamma_\mathcal{H}^c := \frac{c + r_\theta(-s_\theta^*)\left(1 - \frac{\alpha_\theta(-s_\theta^*)}{\alpha_\theta(s_\theta^*)}\right)}{c + r_\theta(s_\theta^*) - r_\theta(-s_\theta^*) \frac{\alpha_\theta(-s_\theta^*)}{\alpha_\theta(s_\theta^*)}} \,.$$

*Assume $\gamma_\mathcal{H} \neq \gamma_\mathcal{H}^c$ and $c < \frac{\alpha_\theta(-s_\theta^*)}{\alpha_\theta(s_\theta^*)} r_\theta(-s_\theta^*)$. The user's strategy at equilibrium is*

$$(f_\theta^{\mathrm{AE}}(s_\theta^*), u_\theta^{\mathrm{AE}}(s_\theta^*)) = (1, 0) \,,$$

$$(f_\theta^{\mathrm{AE}}(-s_\theta^*), u_\theta^{\mathrm{AE}}(-s_\theta^*)) = \begin{cases} \text{any value in } F_\theta \,, & F_\theta \neq \emptyset \,, \\ \left( \frac{\alpha_\theta(-s_\theta^*)}{\alpha_\theta(s_\theta^*)}, 1 \right), & F_\theta = \emptyset \,, \gamma_\mathcal{H} > \gamma_\mathcal{H}^c \,, \\ (1, 0) \,, & F_\theta = \emptyset \,, \gamma_\mathcal{H} < \gamma_\mathcal{H}^c \,. \end{cases}$$

*Proof of Theorem 6.2.* We use a similar notation as in the proof of Theorem 5.1. For improved readability, we drop $\boldsymbol{f}$ and $\boldsymbol{u}$ from $Q_\theta(\boldsymbol{\lambda}, s; \boldsymbol{f}, \boldsymbol{u})$ and $Q_\mathcal{A}(\boldsymbol{\lambda}, s; \boldsymbol{f}, \boldsymbol{u})$. With this notation, the Bellman update for user type $\theta$ in Eq. (11) can be written as

$$Q_\theta(\boldsymbol{\lambda}, s) = f_\theta(s) \, r_\theta(s) - u_\theta(s) \, c$$

$$+ \gamma_\mathcal{H} u_\theta(s) \, f_\theta(s) \, Q_\theta \left( \boldsymbol{\lambda} \hat{\circ} [\boldsymbol{u}(s) \circ \boldsymbol{f}(s)], \arg\max_{s'} Q_\mathcal{A}\left(\boldsymbol{\lambda} \hat{\circ}[\boldsymbol{u}(s) \circ \boldsymbol{f}(s)], s'\right) \right)$$

$$+ \gamma_\mathcal{H} u_\theta(s) \, (1 - f_\theta(s)) \, Q_\theta \left( \boldsymbol{\lambda} \hat{\circ}[\boldsymbol{u}(s) \circ (1 - \boldsymbol{f}(s))], \arg\max_{s'} Q_\mathcal{A}\left(\boldsymbol{\lambda} \hat{\circ}[\boldsymbol{u}(s) \circ (1 - \boldsymbol{f}(s))], s'\right) \right)$$

$$+ \gamma_\mathcal{H} (1 - u_\theta(s)) \, f_\theta(s) \, Q_\theta \left( \boldsymbol{\lambda} \hat{\circ}[(1 - \boldsymbol{u}(s)) \circ \boldsymbol{f}(s)], \arg\max_{s'} Q_\mathcal{A}\left(\boldsymbol{\lambda} \hat{\circ}[(1 - \boldsymbol{u}(s)) \circ \boldsymbol{f}(s)], s'\right) \right)$$

$$+ \gamma_\mathcal{H} (1 - u_\theta(s)) \, (1 - f_\theta(s)) \, Q_\theta \left( \boldsymbol{\lambda} \hat{\circ}[(1 - \boldsymbol{u}(s)) \circ (1 - \boldsymbol{f}(s))], \arg\max_{s'} Q_\mathcal{A}\left(\boldsymbol{\lambda} \hat{\circ}[(1 - \boldsymbol{u}(s)) \circ (1 - \boldsymbol{f}(s))], s'\right) \right).$$

Since the user's entry and subsequent interactions occur under the algorithm's best response, we only need to solve the above for $s \in \arg\max_{s'} Q_{\mathcal{A}}(\boldsymbol{\lambda}, s')$. When $Q_{\mathcal{A}}(\boldsymbol{\lambda}, s) \geq Q_{\mathcal{A}}(\boldsymbol{\lambda}, -s)$, Lemmas C.2, C.5 and C.6 imply

$$(-s) \in \arg\max_{s'} Q_{\mathcal{A}}\big(\boldsymbol{\lambda}\hat{\circ}[\boldsymbol{u}(s) \circ \boldsymbol{f}(s)], s'\big),$$

$$(-s) \in \arg\max_{s'} Q_{\mathcal{A}}\big(\boldsymbol{\lambda}\hat{\circ}[\boldsymbol{u}(s) \circ (1 - \boldsymbol{f}(s))], s'\big),$$

$$s \in \arg\max_{s'} Q_{\mathcal{A}}\big(\boldsymbol{\lambda}\hat{\circ}[(1 - \boldsymbol{u}(s)) \circ \boldsymbol{f}(s)], s'\big),$$

$$(-s) \in \arg\max_{s'} Q_{\mathcal{A}}\big(\boldsymbol{\lambda}\hat{\circ}[(1 - \boldsymbol{u}(s)) \circ (1 - \boldsymbol{f}(s))], s'\big).$$

Plugging these into the Bellman update, we obtain

$$\begin{aligned}
Q_\theta(\boldsymbol{\lambda}, s) = {} & f_\theta(s)\, r_\theta(s) - u_\theta(s)\, c \\
& + \gamma_{\mathcal{H}} u_\theta(s)\, f_\theta(s)\, Q_\theta\big(\boldsymbol{\lambda}\hat{\circ}[\boldsymbol{u}(s) \circ \boldsymbol{f}(s)], -s\big) \\
& + \gamma_{\mathcal{H}} u_\theta(s)\, (1 - f_\theta(s))\, Q_\theta\big(\boldsymbol{\lambda}\hat{\circ}[\boldsymbol{u}(s) \circ (1 - \boldsymbol{f}(s))], -s\big) \\
& + \gamma_{\mathcal{H}}(1 - u_\theta(s))\, f_\theta(s)\, Q_\theta\big(\boldsymbol{\lambda}\hat{\circ}[(1 - \boldsymbol{u}(s)) \circ \boldsymbol{f}(s)], s\big) \\
& + \gamma_{\mathcal{H}}(1 - u_\theta(s))\, (1 - f_\theta(s))\, Q_\theta\big(\boldsymbol{\lambda}\hat{\circ}[(1 - \boldsymbol{u}(s)) \circ (1 - \boldsymbol{f}(s))], -s\big).
\end{aligned}$$

Note that this equation has no dependence on $\boldsymbol{\lambda}$, so, we can drop it from the notation and obtain the following simplified update rule:

$$\begin{aligned}
Q_\theta(s) = {} & f_\theta(s)\, r_\theta(s) - u_\theta(s)\, c \\
& + \gamma_{\mathcal{H}}\big(1 - (1 - u_\theta(s))\, f_\theta(s)\big)\, Q_\theta(-s) \\
& + \gamma_{\mathcal{H}}(1 - u_\theta(s))\, f_\theta(s)\, Q_\theta(s).
\end{aligned}$$

Using Assumptions 1 and 2, we can write the above update separately for $s = s_\theta^*$ and $s = (-s_\theta^*)$:

$$\begin{aligned}
Q_\theta(s_\theta^*) = {} & r_\theta(s_\theta^*) + \gamma_{\mathcal{H}}\, Q_\theta(s_\theta^*), \\
Q_\theta(-s_\theta^*) = {} & f_\theta(-s_\theta^*)\, r_\theta(-s_\theta^*) - u_\theta(-s_\theta^*)\, c \\
& + \gamma_{\mathcal{H}}\big(1 - (1 - u_\theta(-s_\theta^*))\, f_\theta(-s_\theta^*)\big)\, Q_\theta(s_\theta^*) + \gamma_{\mathcal{H}}(1 - u_\theta(-s_\theta^*))\, f_\theta(-s_\theta^*)\, Q_\theta(-s_\theta^*).
\end{aligned}$$

Solving these equations, we obtain

$$\begin{aligned}
Q_\theta(s_\theta^*) = {} & \frac{r_\theta(s_\theta^*)}{1 - \gamma_{\mathcal{H}}}, \\
Q_\theta(-s_\theta^*) = {} & \frac{f_\theta(-s_\theta^*)\, r_\theta(-s_\theta^*) - u_\theta(-s_\theta^*)\, c}{1 - \gamma_{\mathcal{H}}(1 - u_\theta(-s_\theta^*))f_\theta(-s_\theta^*)} + \frac{\gamma_{\mathcal{H}}\big(1 - (1 - u_\theta(-s_\theta^*))f_\theta(-s_\theta^*)\big)\, r_\theta(s_\theta^*)}{\big(1 - \gamma_{\mathcal{H}}(1 - u_\theta(-s_\theta^*))f_\theta(-s_\theta^*)\big)(1 - \gamma_{\mathcal{H}})} \\
= {} & \frac{r_\theta(s_\theta^*)}{1 - \gamma_{\mathcal{H}}} - \frac{r_\theta(s_\theta^*) - f_\theta(-s_\theta^*)\, r_\theta(-s_\theta^*) + u_\theta(-s_\theta^*)\, c}{1 - \gamma_{\mathcal{H}}(1 - u_\theta(-s_\theta^*))f_\theta(-s_\theta^*)}.
\end{aligned}$$

Starting from a prior $\boldsymbol{\lambda}$ over user types, the user's value is

$$V_\theta(\boldsymbol{\lambda}) \coloneqq Q_\theta\big(\arg\max_s Q_{\mathcal{A}}(\boldsymbol{\lambda}, s)\big).$$

Given $V_\theta$, we now explore the user's best strategy that maximizes $V_\theta$, leading to the equilibrium notion defined in Eq. (12). Note that $Q_\theta(s_\theta^*) \geq Q_\theta(-s_\theta^*)$. Therefore, the optimal strategy is to select $(f_\theta(-s_\theta^*), u_\theta(-s_\theta^*))$ such that $s_\theta^* \in \arg\max_s Q_{\mathcal{A}}(\boldsymbol{\lambda}, s)$. In the equilibrium, Theorem 6.1 implies that this is only possible when

$$h_\theta(s_\theta^*)\lambda_\theta = \frac{\lambda_\theta}{1 - \gamma_{\mathcal{A}}(1 - u_\theta(-s_\theta^*))f_\theta(-s_\theta^*)}\left[\frac{1}{\alpha_\theta(s_\theta^*)} - \frac{f_\theta(-s_\theta^*)}{\alpha_\theta(-s_\theta^*)}\right] \geq -\big\langle \boldsymbol{\lambda}_{-\theta}, \boldsymbol{h}_{-\theta}^{\mathrm{AE}}(s_\theta^*)\big\rangle. \quad (24)$$

Here, we generalized $\boldsymbol{h}$ in Eq. (13) by defining $\boldsymbol{h}(s) = \mathbb{1}\{s = a\} \cdot \boldsymbol{h} - \mathbb{1}\{s = b\} \cdot \boldsymbol{h}$. Eq. (24) is a bilinear constraint over $(f_\theta(-s_\theta^*), u_\theta(-s_\theta^*))$:

$$\frac{\lambda_\theta}{\alpha_\theta(s_\theta^*)} + \big\langle \boldsymbol{\lambda}_{-\theta}, \boldsymbol{h}_{-\theta}^{\mathrm{AE}}(s_\theta^*)\big\rangle - f_\theta(-s_\theta^*)\left(\frac{\lambda_\theta}{\alpha_\theta(-s_\theta^*)} + \gamma_{\mathcal{A}}(1 - u_\theta(-s_\theta^*))\big\langle \boldsymbol{\lambda}_{-\theta}, \boldsymbol{h}_{-\theta}^{\mathrm{AE}}(s_\theta^*)\big\rangle\right) \geq 0.$$

Any $(f_\theta(-s_\theta^*), u_\theta(-s_\theta^*))$ that meets this condition is the user's best response. If the condition does not hold, then $V_\theta(\boldsymbol{\lambda}) = Q_\theta(-s_\theta^*)$. In this case, one can verify that for a fixed $u_\theta(-s_\theta^*)$, the sign of the derivative $\frac{\partial Q_\theta(-s_\theta^*)}{\partial f_\theta(-s_\theta^*)}$ does not depend on $f_\theta(-s_\theta^*)$. Similarly, for a fixed $f_\theta(-s_\theta^*)$, the sign of the derivative $\frac{\partial Q_\theta(-s_\theta^*)}{\partial u_\theta(-s_\theta^*)}$ does not depend on $u_\theta(-s_\theta^*)$. Therefore, the optimal strategy is one of the following three edge cases:

$$u_\theta(-s_\theta^*) = 0,\ f_\theta(-s_\theta^*) = 1:\ Q_\theta(-s_\theta^*) = \frac{r_\theta(s_\theta^*)}{1-\gamma_\mathcal{H}} + \frac{r_\theta(-s_\theta^*) - r_\theta(s_\theta^*)}{1-\gamma_\mathcal{H}}$$

$$u_\theta(-s_\theta^*) = 0,\ f_\theta(-s_\theta^*) = 0:\ Q_\theta(-s_\theta^*) = \frac{r_\theta(s_\theta^*)}{1-\gamma_\mathcal{H}} - r_\theta(s_\theta^*)$$

$$u_\theta(-s_\theta^*) = 1,\ f_\theta(-s_\theta^*) \to \frac{\alpha_\theta(-s_\theta^*)}{\alpha_\theta(s_\theta^*)}:\ Q_\theta(-s_\theta^*) = \frac{r_\theta(s_\theta^*)}{1-\gamma_\mathcal{H}} + \frac{\alpha_\theta(-s_\theta^*)}{\alpha_\theta(s_\theta^*)} r_\theta(-s_\theta^*) - r_\theta(s_\theta^*) - c\,.$$

If $c > \frac{\alpha_\theta(-s_\theta^*)}{\alpha_\theta(s_\theta^*)} r_\theta(-s_\theta^*)$, then the third case is dominated by the second case and the problem reduces to the case with no signaling. When $c < \frac{\alpha_\theta(-s_\theta^*)}{\alpha_\theta(s_\theta^*)} r_\theta(-s_\theta^*)$, the user's best response is

$$(u_\theta^{\mathrm{AE}}(-s_\theta^*), f_\theta^{\mathrm{AE}}(-s_\theta^*)) = \begin{cases} \left(1, \to \frac{\alpha_\theta(-s_\theta^*)}{\alpha_\theta(s_\theta^*)}\right), & \gamma_\mathcal{H} > \frac{c + r_\theta(-s_\theta^*)\left(1 - \frac{\alpha_\theta(-s_\theta^*)}{\alpha_\theta(s_\theta^*)}\right)}{c + r_\theta(s_\theta^*) - r_\theta(-s_\theta^*)\frac{\alpha_\theta(-s_\theta^*)}{\alpha_\theta(s_\theta^*)}}\,, \\ (0, 1) & \text{o.w.} \end{cases}$$

Using specific values for type 1 and type 2 users above will complete the proof. $\qquad\square$

**Theorem B.2** (Equilibrium under random entry). *Let $m_\theta^{\mathrm{RE}}$ be the margin of the algorithm's classifier from the perspective of user type $\theta$ when all other user types follow the equilibrium strategy under random entry. When $m_\theta^{\mathrm{RE}} > \lambda_\theta/\alpha_\theta(b)$ and $\gamma_\mathcal{H} \neq r_\theta(a)/r_\theta(b)$ for a user of type $\theta \in \Theta_1$, the user's best strategy is*

$$f_\theta^{\mathrm{RE}}(b) = 1\,, \quad f_\theta^{\mathrm{RE}}(a) = \mathbb{1}\left\{\gamma_\mathcal{H} < \frac{r_\theta(a)}{r_\theta(b)}\right\}.$$

*Proof of Theorem B.2.* We follow a similar notation and conventions as in the proof of Theorem 5.1. We use $Q_\theta^{\mathrm{AE}}$ and $V_\theta^{\mathrm{AE}}$ to denote the Q- and V-value under algorithmic entry. With this notation, the Bellman update for user type $\theta$ in Eq. (2) can be written as

$$Q_\theta(\boldsymbol{\lambda}, s) = f_\theta(s)\, r_\theta(s) + \gamma_\mathcal{H} f_\theta(s)\, V_\theta^{\mathrm{AE}}\big(\boldsymbol{\lambda}\hat{\circ}\boldsymbol{f}(s)\big) + \gamma_\mathcal{H}(1 - f_\theta(s))\, V_\theta^{\mathrm{AE}}\big(\boldsymbol{\lambda}\hat{\circ}(1 - \boldsymbol{f}(s))\big)\,.$$

Recall $V_\theta^{\mathrm{AE}}(\boldsymbol{\lambda}) = Q_\theta^{\mathrm{AE}}\big(\arg\max_s Q_\mathcal{A}(\boldsymbol{\lambda}, s)\big)$. Then, Lemmas C.1 and C.2 imply

$$V_\theta^{\mathrm{AE}}\big(\boldsymbol{\lambda}\hat{\circ}(1 - \boldsymbol{f}(s))\big) = Q_\theta^{\mathrm{AE}}(-s)\,.$$

There remains to determine $V_\theta^{\mathrm{AE}}\big(\boldsymbol{\lambda}\hat{\circ}\boldsymbol{f}(s)\big)$. From the theorem's assumption and using a similar argument as in Corollary 5.2, we can see that regardless of $f_\theta$, when other user types follow the equilibrium strategy, $\langle \boldsymbol{\lambda}, \boldsymbol{h} \rangle \geq 0$. Then, Lemma C.2 implies $\langle \boldsymbol{\lambda}\hat{\circ}\boldsymbol{f}(a), \boldsymbol{h} \rangle \geq 0$, and Lemma C.1 gives

$$V_\theta^{\mathrm{AE}}\big(\boldsymbol{\lambda}\hat{\circ}\boldsymbol{f}(a)\big) = Q_\theta^{\mathrm{AE}}(a)\,.$$

We next consider two possibilities for $V_\theta^{\mathrm{AE}}\big(\boldsymbol{\lambda}\hat{\circ}\boldsymbol{f}(b)\big)$ and show they both yield a similar optimal strategy for $\theta \in \Theta_1$:

- $V_\theta^{\mathrm{AE}}\big(\boldsymbol{\lambda}\hat{\circ}\boldsymbol{f}(b)\big) = Q_\theta^{\mathrm{AE}}(a)$: In this case, we have the following Bellman update:

$$Q_\theta(s) = f_\theta(s)\, r_\theta(s) + \gamma_\mathcal{H} f_\theta(s)\, Q_\theta^{\mathrm{AE}}(a) + \gamma_\mathcal{H}(1 - f_\theta(s))\, Q_\theta^{\mathrm{AE}}(-s)\,.$$

  Note that we have already found $Q_\theta^{\mathrm{AE}}$ in the proof of Theorem 5.1. Now consider a user type $\theta \in \Theta_1$ and a prior $p_1$ over initial content. The user's value is

$$V_\theta = f_\theta(a)\, r_\theta(a)\, p_1(a) + r_\theta(b)\, p_1(b)$$
$$+ \gamma_\mathcal{H}(1 - f_\theta(a))\frac{r_\theta(b)}{1-\gamma_\mathcal{H}} p_1(a)$$
$$+ \gamma_\mathcal{H}(f_\theta(a)\, p_1(a) + p_1(b))\left[\frac{r_\theta(b)}{1-\gamma_\mathcal{H}} - \frac{r_\theta(b) - f_\theta(a)\, r_\theta(a)}{1 - \gamma_\mathcal{H} f_\theta(a)}\right].$$

Calculating $\frac{\partial V_\theta}{\partial f_\theta(a)}$, we find

$$\frac{\partial V_\theta}{\partial f_\theta(a)} = \frac{r_\theta(a) - \gamma_\mathcal{H} r_\theta(b)}{1 - \gamma_\mathcal{H} f_\theta(a)} \left[ p_1(a) + \gamma_\mathcal{H} \frac{f_\theta(a)\, p_1(a) + p_1(b)}{1 - \gamma_\mathcal{H} f_\theta(a)} \right].$$

The term inside the brackets is always positive. Therefore, we can conclude that

$$f_\theta^{\mathrm{RE}}(a) = \begin{cases} 0, & \gamma_\mathcal{H} > \frac{r_\theta(a)}{r_\theta(b)}, \\ 1, & \gamma_\mathcal{H} < \frac{r_\theta(a)}{r_\theta(b)}, \\ \text{any value in } [0,1], & \gamma_\mathcal{H} < \frac{r_\theta(a)}{r_\theta(b)}. \end{cases}$$

- $V_\theta^{\mathrm{AE}}\left(\boldsymbol{\lambda} \hat{\circ} \boldsymbol{f}(b)\right) = Q_\theta^{\mathrm{AE}}(b)$: In this case, we have the following Bellman update:

$$Q_\theta(s) = f_\theta(s)\, r_\theta(s) + \gamma_\mathcal{H} f_\theta(s)\, Q_\theta^{\mathrm{AE}}(s) + \gamma_\mathcal{H}(1 - f_\theta(s))\, Q_\theta^{\mathrm{AE}}(-s).$$

Now consider a user type $\theta \in \Theta_1$ and a prior $p_1$ over initial content. The user's value is

$$V_\theta = f_\theta(a)\, r_\theta(a)\, p_1(a) + r_\theta(b)\, p_1(b)$$
$$+ \gamma_\mathcal{H}\left(p_1(a) - f_\theta(a)\, p_1(a) + p_1(b)\right) \frac{r_\theta(b)}{1 - \gamma_\mathcal{H}}$$
$$+ \gamma_\mathcal{H}\, f_\theta(a)\, p_1(a) \left[ \frac{r_\theta(b)}{1 - \gamma_\mathcal{H}} - \frac{r_\theta(b) - f_\theta(a)\, r_\theta(a)}{1 - \gamma_\mathcal{H} f_\theta(a)} \right].$$

Calculating $\frac{\partial V_\theta}{\partial f_\theta(a)}$, we find

$$\frac{\partial V_\theta}{\partial f_\theta(a)} = \frac{r_\theta(a) - \gamma_\mathcal{H} r_\theta(b)}{1 - \gamma_\mathcal{H} f_\theta(a)} \left[ p_1(a) + \gamma_\mathcal{H} \frac{f_\theta(a)\, p_1(a)}{1 - \gamma_\mathcal{H} f_\theta(a)} \right].$$

The term inside the brackets is always positive. So, $f_\theta(a)^{\mathrm{RE}}$ is similar to the previous case.

$\square$

*Proof of Lemma C.1.* Suppose $\langle \boldsymbol{\lambda}, \boldsymbol{q}(s) \rangle \geq \langle \boldsymbol{\lambda}, \boldsymbol{q}(-s) \rangle$ for some $s$. Lemma C.2 implies

$$\langle \boldsymbol{\lambda}, (\boldsymbol{q}(s) - \boldsymbol{q}(-s)) \circ \boldsymbol{f}(-s) \rangle = -\langle \boldsymbol{\lambda} \circ \boldsymbol{f}(-s), \boldsymbol{q}(-s) - \boldsymbol{q}(s) \rangle$$
$$\leq -\langle \boldsymbol{\lambda}, \boldsymbol{q}(-s) - \boldsymbol{q}(s) \rangle = \langle \boldsymbol{\lambda}, \boldsymbol{q}(s) - \boldsymbol{q}(-s) \rangle.$$

Plugging this into $Q_\mathcal{A}(\boldsymbol{\lambda}, -s)$, as defined in Eq. (15), yields

$$Q_\mathcal{A}(\boldsymbol{\lambda}, -s) \leq \langle \boldsymbol{\lambda}, \boldsymbol{q}(-s) \rangle + \gamma_\mathcal{A} \langle \boldsymbol{\lambda}, \boldsymbol{q}(s) - \boldsymbol{q}(-s) \rangle$$
$$\leq \langle \boldsymbol{\lambda}, \boldsymbol{q}(-s) \rangle + \gamma_\mathcal{A} \langle \boldsymbol{\lambda}, \boldsymbol{q}(s) - \boldsymbol{q}(-s) \rangle + (1 - \gamma_\mathcal{A}) \langle \boldsymbol{\lambda}, \boldsymbol{q}(s) - \boldsymbol{q}(-s) \rangle$$
$$= \langle \boldsymbol{\lambda}, \boldsymbol{q}(s) \rangle \leq Q_\mathcal{A}(\boldsymbol{\lambda}, s).$$

This completes the proof. $\square$

*Proof of Lemma C.2.* We first find a simplified expression for $\boldsymbol{q}(s) - \boldsymbol{q}(-s)$. Using the definition in Eq. (16), a straightforward calculation gives

$$\boldsymbol{q}(s) - \boldsymbol{q}(-s) = \frac{1 - \gamma_\mathcal{A}}{(1 - \gamma_\mathcal{A} \boldsymbol{f}(s)) \circ (1 - \gamma_\mathcal{A} \boldsymbol{f}(-s))} \circ \left[ \boldsymbol{f}(s) \circ \frac{1}{\boldsymbol{\alpha}(s)} - \boldsymbol{f}(-s) \circ \frac{1}{\boldsymbol{\alpha}(-s)} \right]. \quad (25)$$

For a content $s$, under Assumption 1, if $s = s_\theta^*$, then $f_\theta(s) = 1$. Otherwise, either $f_\theta(s) = 1$ or $f_\theta(s) < \alpha_\theta(s)/\alpha_\theta(-s)$. Therefore, we can divide $\Theta$ into two groups where in one group $f_\theta(s) = 1$ and in the other group $f_\theta(s) < \alpha_\theta(s)/\alpha_\theta(-s)$ and $f_\theta(-s) = 1$. Using this, we have

$$\langle \boldsymbol{\lambda} \circ \boldsymbol{f}(s), \boldsymbol{q}(s) - \boldsymbol{q}(-s) \rangle = \sum_{\theta \in \Theta} \lambda_\theta f_\theta(s)(q_\theta(s) - q_\theta(-s))$$
$$= \sum_{\theta: f_\theta(s) < \frac{\alpha_\theta(s)}{\alpha_\theta(-s)}} \lambda_\theta f_\theta(s)(q_\theta(s) - q_\theta(-s)) + \sum_{\theta: f_\theta(s) = 1} \lambda_\theta (q_\theta(s) - q_\theta(-s)).$$

For the first group corresponding to the first sum above, Eq. (25) implies that

$$\text{sign}\left(q_\theta(s) - q_\theta(-s)\right) = \text{sign}\left(f_\theta(s)\frac{1}{\alpha_\theta(s)} - \frac{1}{\alpha_\theta(-s)}\right) = -1\,.$$

Therefore, we can conclude

$$\langle \boldsymbol{\lambda} \circ \boldsymbol{f}(s), \boldsymbol{q}(s) - \boldsymbol{q}(-s)\rangle \geq \langle \boldsymbol{\lambda}, \boldsymbol{q}(s) - \boldsymbol{q}(-s)\rangle\,.$$

$\square$

*Proof of Lemma C.3.* Suppose $\langle \boldsymbol{\lambda}, \boldsymbol{q}(s)\rangle \geq \langle \boldsymbol{\lambda}, \boldsymbol{q}(-s)\rangle$ for some $s$. Lemma C.2 implies

$$\begin{aligned}
\langle \boldsymbol{\lambda}, (\boldsymbol{q}(s) - \boldsymbol{q}(-s)) \circ \boldsymbol{f}(s)\rangle &= \langle \boldsymbol{\lambda} \circ \boldsymbol{f}(s), \boldsymbol{q}(s) - \boldsymbol{q}(-s)\rangle \\
&\geq \langle \boldsymbol{\lambda}, \boldsymbol{q}(s) - \boldsymbol{q}(-s)\rangle \geq 0\,.
\end{aligned}$$

Plugging this into $Q_{\mathcal{A}}(\boldsymbol{\lambda}, s)$, as defined in Eq. (15), yields

$$Q_{\mathcal{A}}(\boldsymbol{\lambda}, s) = \langle \boldsymbol{\lambda}, \boldsymbol{q}(s)\rangle\,.$$

$\square$

*Proof of Lemma C.4.* Assuming $\langle \boldsymbol{\lambda}, 1 - \boldsymbol{f}(s)\rangle > 0$, Lemma C.2 implies

$$\langle \boldsymbol{\lambda} \circ (1 - \boldsymbol{f}(s)), \boldsymbol{q}(s) - \boldsymbol{q}(-s)\rangle \leq 0 \iff \langle \boldsymbol{\lambda}\hat{\circ}(1 - \boldsymbol{f}(s)), \boldsymbol{q}(s) - \boldsymbol{q}(-s)\rangle \leq 0\,.$$

Then Lemma C.1 implies

$$\arg\max_{s'} Q_{\mathcal{A}}\left(\boldsymbol{\lambda}\hat{\circ}(1 - \boldsymbol{f}(s)), s'\right) = (-s)\,.$$

Finally, Lemma C.3 implies

$$\max_{s'} Q_{\mathcal{A}}\left(\boldsymbol{\lambda}\hat{\circ}(1 - \boldsymbol{f}(s)), s'\right) = \langle \boldsymbol{\lambda}\hat{\circ}(1 - \boldsymbol{f}(s)), \boldsymbol{q}(-s)\rangle\,.$$

$\square$

*Proof of Lemma C.5.* Suppose $\langle \boldsymbol{\lambda}, \boldsymbol{q}(s)\rangle \geq \langle \boldsymbol{\lambda}, \boldsymbol{q}(-s)\rangle$ for some $s$. Lemmas C.2 and C.6 imply

$$\begin{aligned}
\langle \boldsymbol{\lambda}, (\boldsymbol{q}(s) - \boldsymbol{q}(-s)) \circ (1 - \boldsymbol{u}(-s)) \circ \boldsymbol{f}(-s)\rangle &= -\langle \boldsymbol{\lambda} \circ (1 - \boldsymbol{u}(-s)) \circ \boldsymbol{f}(-s), \boldsymbol{q}(-s) - \boldsymbol{q}(s)\rangle \\
&\leq -\langle \boldsymbol{\lambda}, \boldsymbol{q}(-s) - \boldsymbol{q}(s)\rangle = \langle \boldsymbol{\lambda}, \boldsymbol{q}(s) - \boldsymbol{q}(-s)\rangle\,.
\end{aligned}$$

Plugging this into $Q_{\mathcal{A}}(\boldsymbol{\lambda}, -s)$, as defined in Eq. (20), yields

$$\begin{aligned}
Q_{\mathcal{A}}(\boldsymbol{\lambda}, -s) &\leq \langle \boldsymbol{\lambda}, \boldsymbol{q}(-s)\rangle + \gamma_{\mathcal{A}}\langle \boldsymbol{\lambda}, \boldsymbol{q}(s) - \boldsymbol{q}(-s)\rangle \\
&\leq \langle \boldsymbol{\lambda}, \boldsymbol{q}(-s)\rangle + \gamma_{\mathcal{A}}\langle \boldsymbol{\lambda}, \boldsymbol{q}(s) - \boldsymbol{q}(-s)\rangle + (1 - \gamma_{\mathcal{A}})\langle \boldsymbol{\lambda}, \boldsymbol{q}(s) - \boldsymbol{q}(-s)\rangle \\
&= \langle \boldsymbol{\lambda}, \boldsymbol{q}(s)\rangle \leq Q_{\mathcal{A}}(\boldsymbol{\lambda}, s)\,.
\end{aligned}$$

This completes the proof. $\square$

*Proof of Lemma C.6.* We first find a simplified expression for $\boldsymbol{q}(s) - \boldsymbol{q}(-s)$. Using the definition in Eq. (21) and $\boldsymbol{u}(s) \circ \boldsymbol{u}(-s) = 0$ and $\boldsymbol{u}(s) \circ (1 - \boldsymbol{f}(-s)) = 0$ as implied by Assumption 2, a straightforward calculation gives

$$\boldsymbol{q}(s) - \boldsymbol{q}(-s) = \frac{1 - \gamma_{\mathcal{A}}}{(1 - \gamma_{\mathcal{A}}(1 - \boldsymbol{u}(s)) \circ \boldsymbol{f}(s)) \circ (1 - \gamma_{\mathcal{A}}(1 - \boldsymbol{u}(-s)) \circ \boldsymbol{f}(-s))} \circ \left[\boldsymbol{f}(s) \circ \frac{1}{\boldsymbol{\alpha}(s)} - \boldsymbol{f}(-s) \circ \frac{1}{\boldsymbol{\alpha}(-s)}\right]\,.$$

(26)

For a content $s$, under Assumption 2, whenever $u_\theta(s) > 0$, we have $f_\theta(s) < \alpha_\theta(s)/\alpha_\theta(-s)$. Therefore, we can divide $\Theta$ into two groups where in one group $u_\theta(s) = 0$ and in the other group $u_\theta(s) > 0, f_\theta(s) < \alpha_\theta(s)/\alpha_\theta(-s)$:

$$\begin{aligned}
\langle \boldsymbol{\lambda} \circ (1 - \boldsymbol{u}(s)), \boldsymbol{q}(s) - \boldsymbol{q}(-s)\rangle &= \sum_{\theta \in \Theta} \lambda_\theta(1 - u_\theta(s))(q_\theta(s) - q_\theta(-s)) \\
&= \sum_{\theta : u_\theta(s) > 0} \lambda_\theta(1 - u_\theta(s))(q_\theta(s) - q_\theta(-s)) + \sum_{\theta : u_\theta(s) = 0} \lambda_\theta(q_\theta(s) - q_\theta(-s))\,.
\end{aligned}$$

For the first group corresponding to the first sum above, we know from Assumption 2 that $s = (-s_\theta^*)$. In this case, Assumption 1 implies $f_\theta(-s) = 1$. Plugging this into Eq. (25), we have

$$\text{sign}\left(q_\theta(s) - q_\theta(-s)\right) = \text{sign}\left(f_\theta(s)\frac{1}{\alpha_\theta(s)} - \frac{1}{\alpha_\theta(-s)}\right) = -1\,.$$

Therefore, we can conclude

$$\langle \boldsymbol{\lambda} \circ (1 - \boldsymbol{u}(s)), \boldsymbol{q}(s) - \boldsymbol{q}(-s)\rangle \geq \langle \boldsymbol{\lambda}, \boldsymbol{q}(s) - \boldsymbol{q}(-s)\rangle\,.$$

$\square$

*Proof of Lemma C.7.* Suppose $\langle \boldsymbol{\lambda}, \boldsymbol{q}(s)\rangle \geq \langle \boldsymbol{\lambda}, \boldsymbol{q}(-s)\rangle$ for some $s$. Lemmas C.2 and C.6 imply

$$\langle \boldsymbol{\lambda}, (\boldsymbol{q}(s) - \boldsymbol{q}(-s)) \circ (1 - \boldsymbol{u}(s)) \circ \boldsymbol{f}(s)\rangle = \langle \boldsymbol{\lambda} \circ (1 - \boldsymbol{u}(s)) \circ \boldsymbol{f}(s), \boldsymbol{q}(s) - \boldsymbol{q}(-s)\rangle$$
$$\geq \langle \boldsymbol{\lambda}, \boldsymbol{q}(s) - \boldsymbol{q}(-s)\rangle \geq 0\,.$$

Plugging this into $Q_\mathcal{A}(\boldsymbol{\lambda}, s)$, as defined in Eq. (20), yields

$$Q_\mathcal{A}(\boldsymbol{\lambda}, s) = \langle \boldsymbol{\lambda}, \boldsymbol{q}(s)\rangle\,.$$

$\square$

*Proof of Lemma C.8.* Assuming $\langle \boldsymbol{\lambda}, \boldsymbol{u}(s) \circ \boldsymbol{g}(s)\rangle > 0$, Lemma C.6 implies

$$\langle \boldsymbol{\lambda} \circ \boldsymbol{g}(s) \circ \boldsymbol{u}(s), \boldsymbol{q}(s) - \boldsymbol{q}(-s)\rangle \leq 0 \iff \langle \boldsymbol{\lambda}\hat{\circ}[\boldsymbol{u}(s) \circ \boldsymbol{g}(s)], \boldsymbol{q}(s) - \boldsymbol{q}(-s)\rangle \leq 0\,.$$

Then Lemma C.5 implies

$$\arg\max_{s'} Q_\mathcal{A}\left(\boldsymbol{\lambda}\hat{\circ}[\boldsymbol{u}(s) \circ \boldsymbol{g}(s)], s'\right) = (-s)\,.$$

Finally, Lemma C.7 implies

$$\max_{s'} Q_\mathcal{A}\left(\boldsymbol{\lambda}\hat{\circ}[\boldsymbol{u}(s) \circ \boldsymbol{g}(s)], s'\right) = \langle \boldsymbol{\lambda}\hat{\circ}[\boldsymbol{u}(s) \circ \boldsymbol{g}(s)], \boldsymbol{q}(-s)\rangle\,.$$

$\square$

