# OpenReview forum: "The Burden of Interactive Alignment with Inconsistent Preferences"
_NeurIPS.cc/2025/Conference — NeurIPS 2025 poster_

### Official Review · Reviewer_HcaA · 2025-07-01

**Clarity:** 3
**Significance:** 3
**Originality:** 3
**Rating:** 5
**Confidence:** 3

**Summary:**

In this paper, the authors introduce a model to study how users with inconsistent preferences (i.e., choices don’t always represent true preferences) can align algorithms with their true interests. The interaction is modeled as a Stackelberg game where the user is the leader, committing to a strategy, and the algorithm is the follower. The user has a “system 2” for engagement decisions and a “system 1” for the duration of such engagement, while the algorithm maximizes engagement. They define the "burden of alignment" as the level of foresight a user needs to steer the algorithm effectively (horizon). Their theoretical results show that foresighted users can improve alignment, which can be challenging, and the burden can be reduced with a costly signal for the user (e.g., clicks to show disinterest). Otherwise, myopic users align with the algorithm’s objective.

**Questions:**

- Line 167: As a minor notation detail, it would be helpful for clarity to define acronyms for random entry (RE) and algorithmic entry (AE).
- Line 302: Something is wrong with this sentence.

**Ethical Concerns:**

["NO or VERY MINOR ethics concerns only"]

**Final Justification:**

I think this paper is great and should be accepted. The authors study an important and very interesting problem: a scenario of inconsistent preferences when users deal with an engagement-maximizing algorithm. They define the "burden of alignment" as the level of foresight a user needs to steer the algorithm effectively (horizon). Their theoretical results show that foresighted users can improve alignment, which can be challenging, and the burden can be reduced with a costly signal for the user (e.g., clicks to show disinterest)

I didn't find any weaknesses when reviewing the paper, but I had some suggestions that the authors addressed during the discussion period.

**Limitations:**

Yes, in the Discussion section, and other assumptions throughout the manuscript

**Quality:**

4

**Strengths And Weaknesses:**

This is a great paper. It offers a rigorous theoretical framework of interactive alignment by modeling the user-algorithm dynamic as a Stackelberg game with the user as a leader. Its primary strength lies in departing from the simplifying assumption of user rationality and instead focusing on what users can do when they carry the burden of alignment by interacting with an engagement-maximizing algorithm.

It provides theoretical guarantees under well-defined assumptions: abstracting the algorithm’s learning process, system 2 decides engagement and system 1 its duration, the algorithm maximizes engagement, users are aware of their rewards, the algorithm has full knowledge of strategies, etc.

While the paper frames the problem of inconsistent preferences in general terms, this same setup applies to the real phenomenon of doomscrolling. I’m not pointing this out as a limitation or a weakness, but maybe as something to consider. This theoretical framework could be used to study this effect—from the user and algorithm perspective—to improve users’ well-being. I see that as a strength.

In the paper, the user is modeled by their type $\theta \in \Theta$ for a specific session, allowing the same user to be of different types across sessions. However, the user’s foresight $\gamma_H$ is fixed, and it might be interesting to consider a dynamic discounting factor for a user. In reality, a person's ability to be foresighted can fluctuate based on internal factors. For example, a user who is typically strategic might enter a myopic state (doomscrolling) for some sessions. I wonder what the burden of alignment is in this situation. Would it degrade over time in this tug of war, where your myopic sessions ruin your strategic ones? Would the costly signals be able to reduce the burden? What happens when a lot of users doomscroll at the same time (sending a strong signal to the algorithm) during moments of political turmoil or catastrophes? I could also see how a user can resist the temptation of item `a` but not if this item is shown repeatedly.

---

> ### Author Rebuttal · Authors · 2025-07-31
>
> We thank the reviewer for their thorough and positive review of our work. We’re truly heartened that the reviewer described the paper as “great,” and appreciated its rigorous theoretical framework and guarantees under well-defined assumptions. We’re especially pleased that the reviewer was able to draw connections to real-world phenomena and raised thoughtful questions.
>
> > ... it might be interesting to consider a dynamic discounting factor for a user. In reality, a person's ability to be foresighted can fluctuate based on internal factors. For example, a user who is typically strategic might enter a myopic state (doomscrolling) for some sessions. I wonder what the burden of alignment is in this situation. Would it degrade over time in this tug of war, where your myopic sessions ruin your strategic ones? Would the costly signals be able to reduce the burden? What happens when a lot of users doomscroll at the same time (sending a strong signal to the algorithm) during moments of political turmoil or catastrophes?
>
> These are great questions! While we can offer some intuitive answers within the scope of this rebuttal, several of these points suggest promising directions for future work. They also highlight how our framework—and its departure from standard rationality assumptions in alignment—can raise and help address new, nuanced questions.
>
> Regarding fluctuating user’s discount factor, for simplicity, first suppose $\gamma_{\mathcal{H}}$ is fixed across users but can vary over time for a given user. In this case, the critical threshold for alignment remains mostly as before; the key question becomes whether this user’s $\gamma_{\mathcal{H}}$ exceeds this threshold at any given time. Now, consider a scenario in which $\gamma_{\mathcal{H}}$ of the user decreases when the user is relaxing or is a "type 2" user in our special case. Since misalignment is minimal in such contexts, fluctuations in patience are unlikely to affect outcomes significantly. However, if the user becomes more myopic precisely when exposed to tempting but misaligned content—as in the case of "type 1" users—their effective horizon may become too short to steer the algorithm effectively.
>
> In more complex settings where the $\gamma_{\mathcal{H}}$ values of many users change simultaneously, externalities could arise, affecting the critical effective horizon required for alignment. This is a particularly intriguing question and a compelling direction for future research.
>
> > Re. Line 167 and 302
>
> We apologize for the typo in line 302. The sentence should read: “Given the algorithm’s best response, we now formally characterize the equilibria under algorithmic entry when users…” We will also clarify the acronyms RE and AE at this point in the text. We thank the reviewer for their careful reading and for pointing out these issues.

---

### Official Review · Reviewer_8uFD · 2025-07-02

**Clarity:** 2
**Significance:** 3
**Originality:** 2
**Rating:** 3
**Confidence:** 2

**Summary:**

The authors study an Nash equilibrium of a game where two parties,
(recommendation) algorithm and user, interact.  In this setting algorithm
recommends content, which user then may be engage for certain time. The user is
then rewarded afterwards.  The objective of the algorithm is to maximize the
engagement of the user, whereas the goal of each user is to maximize the
reward.

**Questions:**

- Are these results (easily) extendable to n items?
- What are the practical implications for recommendation engines?

**Ethical Concerns:**

["NO or VERY MINOR ethics concerns only"]

**Final Justification:**

I have read the comments provided by the authors. The theoretical contributions are sufficient for publication.
The major downside of the paper is the lack of experiments, and because of that it is not clear what are the exact practical implications of the authors results. Some set of experiments providing some connections of the theoretical results to applications would have alleviated these concerns.

**Limitations:**

yes

**Paper Formatting Concerns:**

no issues

**Quality:**

3

**Strengths And Weaknesses:**

+ Interesting results regarding the equilibrium, especially with inconsistent rewards. These contributions advance the field, and are sufficient for a publication.

- The contributions are all theoretical. The main downside with the paper is that it does not have any experiments or any use case study. It is not clear what are the practical implications of these results. Some experiments or a small use case study would alleviate these issues.
- The authors motivate their example using recommendation algorithms. Yet the related work
  in that domain is not thoroughly discussed. The authors should have discussed the recommendation algorithm in the related work more thoroughly.
- The paper focuses on cases with two items. The results should be extendable,
  so it would be helpful to see a more general results, especially for Theorem 4.1.

---

> ### Author Rebuttal · Authors · 2025-07-31
>
> We thank the reviewer for taking the time to read and review our work. We’re glad they found our results interesting and hope they find our responses to their concerns below to be convincing.
>
> > the contributions are all theoretical … what are the practical implications for recommendation engines?
>
> As Dean et al. put it, we believe that formal mathematical models like ours can offer valuable insights and help guide empirical research—particularly in evaluating and motivating alternative designs, such as our analysis of costly signaling. We fully agree with the reviewer that theoretical results, no matter how strong, are only part of the answer to broader policy questions.
>
> In terms of implications, beyond formalizing the burden that alignment places on users with inconsistent preferences, our findings on how costly signaling can alleviate this burden have significant practical relevance. As our special case illustrates, platforms that maximize engagement length often have no incentive—and may even have a disincentive—to introduce friction or support user signaling. This is reflected in the widespread shift toward frictionless scrolling interfaces on video platforms.
>
> However, our framework suggests that introducing diverse options—even those that don’t directly affect engagement but instead act as costly signals of user intent (such as a “show less like this” button, as suggested by Reviewer 7AFQ)—can help reduce the alignment burden. Our work also raises additional practical questions, such as how middleware tools (as discussed by Fukuyama et al.) might be used to extend users' effective horizons when their own patience is insufficient for alignment. We thank the reviewer for raising this important point and will be sure to expand on these connections in the updated manuscript.
>
> Fukuyama, et al. (2020). Report of the working group on platform scale. Program on Democracy and the Internet, Stanford University
>
> > The authors motivate their example using recommendation algorithms. Yet the related work in that domain is not thoroughly discussed.
>
> We’re sorry to hear that the reviewer found our related work section insufficient. We would like to clarify that our focus has primarily been on recent theoretical modeling efforts in this space, including works by Haupt et al., Kleinberg et al., Zhao et al., Haghtalab et al., and Hajiaghayi et al. While space constraints limited what we could include in the main text, we have done our best to expand on additional relevant work in Appendix A. We would greatly welcome any suggestions the reviewer may have for further enriching this section.
>
> >  Are these results (easily) extendable? … it would be helpful to see a more general results, especially for Theorem 4.1
>
> The abstraction in Sec. 3.4 and then Thm 4.1 may be more general than it appears: in many settings, item types can be interpreted broadly, and with reasonable approximations, users may consistently benefit more from one type over another, more tempting alternative. We believe our closed-form results here likely extend beyond this special case, as this structure often arises as a component of more complex patterns encountered in practice. That said, we currently do not have formal results beyond simulation evidence, and we expect that the theoretical results may become less elegant or interpretable outside this simplified setting.

---

### Official Review · Reviewer_bpFx · 2025-07-02

**Clarity:** 3
**Significance:** 3
**Originality:** 3
**Rating:** 4
**Confidence:** 4

**Summary:**

This paper studies a setting where humans sometimes act in ways that are systematically at odds with their true preferences (following the system 1 system 2 model in Kleinberg & Raghavan). An algorithm that aims to maximize engagement is attempting to suggest content to a human, where some humans have preferences aligned with the algorithm’s prior, and some have preferences at odds with it. This paper studies when humans can effectively strategize to end up being suggested the content that they prefer.

**Questions:**

Highlighted in the "weaknesses" section - in particular, my questions about the meaning of the user being more or less myopic, and the policy implications.

**Ethical Concerns:**

["NO or VERY MINOR ethics concerns only"]

**Final Justification:**

I enjoyed reading this paper, and during the final discussion the authors convinced me that the time horizon issue I had brought up is less relevant. I was on the fence about raising my score (if I could raise it a half-unit I likely would).

**Limitations:**

Yes - but more discussion (and connection beyond theory) could be helpful.

**Quality:**

4

**Strengths And Weaknesses:**

Strengths:
This paper studies an interesting problem, and ties its work to prior work both in practice(Haupt et al and Cen et al) and theory (Kleinberg & Raghavan). I appreciated that the paper included several extensions (e.g. including ‘costly signaling’, which was a variant that I hadn’t immediately thought of).

Weaknesses:
A few parts of the model felt unclear. For example, one crucial factor is the human’s time horizon, which influences their strategy as it influences how long they can take the costly action of signaling a different preference: in section 5, it is shown that if a user has a low enough time horizon, then they may incur relatively heavy regret. However, in real-life settings, users may have relatively short time horizons precisely because they know they will not be using the app for very long: for example, if a user installs an app that they know they will only use for a short period of time (e.g. in a country that they infrequently visit), then they may (rationally) decide that they don’t want to invest the effort in steering what the app may suggest to them. By the results in this paper, this would imply a *high* regret, but in reality this would likely lead to a low regret. In this paper, I believe the authors are focusing on the case where a consumer’s time horizon is driven solely by how much effort they wish to put in (or how myopic they are), rather than any actual desire to leave the platform. While I don’t think this substantially changes the core of the paper, it would be useful for the paper to note the different reasons why users may have different time horizons.

As one other critique, the paper is very theory-heavy - which I appreciate, as its main goal is clearly a theoretical understanding. However, I would have appreciated more explicit connection to practice or policy, even in the appendix. For example, what would the results in Section 6 imply about how algorithms should be designed? How would the results change if algorithms have different incentives beyond engagement (e.g. different values for items)?

---

> ### Author Rebuttal · Authors · 2025-07-31
>
> We thank the reviewer for their insightful review of our work. We’re glad they found the problem interesting and appreciated how it connects prior theoretical and empirical research. We’re also pleased that the reviewer recognized the value of our proposed extensions. Below, we address the reviewer’s questions and concerns.
>
> >  it is shown that if a user has a low enough time horizon, then they may incur relatively heavy regret. However, in real-life settings, users may have relatively short time horizons precisely because they know they will not be using the app for very long. … I believe the authors are focusing on the case where a consumer’s time horizon is driven solely by how much effort they wish to put in, rather than any actual desire to leave the platform.
>
> The reviewer’s interpretation is correct, and we would like to use this example as an opportunity to further clarify our model. When calculating regret, we consider a session that can last arbitrarily long, where the user has no intention of leaving the platform but is instead strategizing to maximize the value of the interaction. Our notion of a discount factor captures how patient the user is across these interactions or over what horizon they are capable of optimizing. As the reviewer pointed out, this is not the horizon that the user might intend to leave. We agree that if the user is aware of an early cutoff time within the session, they may prioritize immediate consumption over long-term optimization. However, if this cutoff is sufficiently far in the future—such as planning to delete the app a year from now—we expect our results to apply readily.
>
> > The paper is very theory-heavy - which I appreciate, as its main goal is clearly a theoretical understanding. However, I would have appreciated more explicit connection to practice or policy, even in the appendix. For example, what would the results in Section 6 imply about how algorithms should be designed?
>
> Our goal in this theoretical paper is to provide a first-of-its-kind analysis of aligning with inconsistent preferences and the burden this places on strategic users. We have intentionally presented the model and results in abstract yet interpretable terms to highlight the generality of the framework and its potential applications. That said, we agree that additional context and stronger connections to practice would be helpful. To that end, we have included concrete examples in our response to Reviewer 7AFQ, which we hope the reviewer will also find useful.
>
> Regarding the results in the presence of costly signaling (Sec 6), we show that giving users just one additional option—even one that simply burns effort—can significantly improve their ability to steer the algorithm. As noted by Reviewer 7AFQ, a small "show less like this" button, like those found on older platforms, can serve this purpose. Such costly actions, while not directly altering engagement with the content, send a strong signal to the platform and can ease the burden of alignment on the user.
>
> From a policy perspective, our special case already illustrates that engagement maximization alone is not sufficient to motivate platforms to offer diverse options to users. In fact, the incentives often run in the opposite direction: platforms may reduce friction as much as possible, leading to the kind of seamless, mindless scrolling now ubiquitous in video recommenders. Of course, these are speculative insights derived from our model. Strong empirical work is needed to bridge theory and policy. As Dean et al. put it, we hope that formal mathematical models like ours can guide and inspire future empirical research by offering valuable insights—while recognizing that they do not provide final answers.
>
> > How would the results change if algorithms have different incentives beyond engagement?
>
> This is a great question. Our theoretical framework does in fact accommodate this case. One simply needs to define $1/\alpha_\theta(s)$ as the expected value—or more generally, whatever measure of utility—the platform assigns to content $s$. The rest of our results then apply directly. We chose to define $1/\alpha_\theta(s)$ as expected engagement length in our exposition to simplify the presentation, as many platforms monetize based on engagement duration. However, our results are more general and extend beyond this specific interpretation.

---

> > ### Comment · Reviewer_bpFx · 2025-08-03
> >
> > Thank you for your response! I appreciate your thoughtful discussion of my questions.

---

### Official Review · Reviewer_7AFQ · 2025-07-03

**Clarity:** 4
**Significance:** 2
**Originality:** 3
**Rating:** 5
**Confidence:** 4

**Summary:**

Recommendation algorithms learn user preferences using user behavior, which may be rational (System 2) or short-sighted/greedy (System 1). Due to the greedy behavior, the algorithms may be unable to distinguish between users whose rational selves agree/disagree with their greedy selves. To address this, users must strategically choose their rational behavior so as to signal their type to the algorithm. This paper develops a theoretical model of this setting as a Stackleberg game, and analyzes Bellman equations to show that (a) users may need to substantially sacrifice / alter their behavior in order to "steer" the algorithm, (b) users who highly discount future rewards will be unable to steer the algorithm at all, and (c) a costly signal may mitigate the burden to users.

**Questions:**

1. What recommendation setting motivates this work?

2. How restrictive the special case in 3.4? Does it capture most settings? Do you have a sense of how the results extend (steering becomes easier/harder) beyond this special case?

3. Is a "show less like this" button seen on a few platforms currently an example of a costly signal? What would motivate a platforms to implement a costly signaling feature-- is there a burden on the platform as well?

4. Modeling questions:
- On a session-level, why would the algorithm discount future rewards?
- When computing the algorithm's Q-value, why does the algorithm assume the users will continue to use the same strategy $f$ in the future?
- Is pulling the expected duration into the reward $r_\theta$ (in line 120) really without loss of generality? Does it result in some additional dependence on $\gamma_H$?
- What does "when users have consistent preferences ... the alignment problem reduces to engagement maximization" mean on page 1? My intuition is that there is misalignment even with only System 2 (e.g. some content could have a reward only after time $T$ has been reached, like a suspenseful movie, and the algorithm is motivated to promote that content). Is this misalignment possible in general or only because I chose pathological rewards? If it can occur in general, do you have a sense of how hard it is for a user to steer the algorithm without inconsistent preferences (as a baseline)?

**Ethical Concerns:**

["NO or VERY MINOR ethics concerns only"]

**Final Justification:**

I will maintain my positive score. The authors' rebuttal has addressed my questions.

**Limitations:**

Yes

**Quality:**

4

**Strengths And Weaknesses:**

I enjoyed reading this paper; the paper was well organized, and the authors provided clear and intuitive interpretations of the theoretical results, which appear non-trivial. Recent literature has examined both strategic and inconsistent content consumption, but this is a novel bridge between these lines of work. The findings are also interesting - users may potentially require great energy and foresightedness to align the algorithm due to inconsistent preferences, and sometimes it is in fact impossible to align.

One weakness is that it's not clear what setting motivates this work: in what type of content recommendation does this phenomenon (users strategically determining which content to consume, but then greedily determining how long to consume it for) occur? In particular: in what settings/platforms does a user consume a *particular* piece of content for an undefined length, and the platform wants to maximize this length of time.

Moreover, while the model is quite general, the results are all in a special case where there are two items.

I also have a few quibbles/clarifications about the modeling decisions, which I ask about below. It would also be interesting to have additional interpretable insights from the model, such as discussion of which types of users the burden of alignment is highest for (with and without costly signaling).

---

> ### Author Rebuttal · Authors · 2025-07-31
>
> We thank the reviewer for their thorough and insightful feedback. We couldn’t have asked for a more engaged and in-depth review of our work. We are especially pleased that the reviewer found the paper well-organized and our interpretations clear and intuitive. Below, we address the specific questions raised.
>
> > What recommendation setting motivates this work?
>
> We have intentionally presented our setting in an abstract form because we believe it encompasses a wide range of applications—an observation echoed by reviewers, to our excitement. To provide a more concrete illustration and clarify the model’s assumptions and generality, we begin with a simple example of the interaction structure, followed by a discussion of how our user decision-making model builds on prior work. We apologize in advance as this response will be on the long side.
>
> Consider the following example: a user opens a music or video recommender system with a specific intent. Upon seeing a recommendation—say, a country song—the user decides whether to engage (i.e., listen) or to skip and disengage for a while. Once the user chooses to engage, however, they may relinquish rational control over how long they remain engaged, with the actual duration influenced by the content’s emotional or addictive pull.
> Suppose this user is working while listening and would benefit most from calm music. However, they are a fan of a particular artist, X, whose songs may be more distracting. The platform profits from longer engagement—via ad revenue or increased interaction—and therefore has an incentive to repeatedly recommend content like X’s music. In this case, the user exhibits inconsistent preferences: they prefer calm music for productivity, but are tempted by content that undermines that goal. This type of user falls under our "type 1" users, where the reward and engagement duration are misaligned across content types (here, X’s music vs. calm music). Notably, in a different context—say, when the user is relaxing rather than working—their preferences may align with the platform’s and they are "type 2" users. This motivates our session-based modeling of interactions.
>
> Our framework extends beyond recommender systems. For instance, a chatbot that charges per API call may operate in various "modes." In an educational mode, longer sessions may be valuable for a student, aligning incentives. But for an engineer seeking a quick answer, a shorter session is preferable. Similarly, a therapy chatbot operating in an "affirmative" mode might prolong interaction by being emotionally validating, even if it does not help the user confront deeper issues. These examples all fit our model, provided we flexibly define user types and item spaces, even approximately, and work with a tractable number of them.
>
> Turning to our user model: there is ample empirical support for user strategizing. For example, Cem et al. observed that "[some users] ignore content they actually like to" in order to shape future recommendations. Likewise, Haupt et al. found that users "explicitly and actively curate their feeds in order to influence the platform to serve them content of a *particular type* in the future." These findings indicate that users engage in strategic behavior.
>
> Our user model—where users strategically choose whether to engage with content, but then greedily decide how long to engage—is inspired by the seminal work of Kleinberg et al. As they note, human preferences are inherently inconsistent and psychologically rich, but their "two-mind" model offers a tractable abstraction. In this model, System 2 represents the long-run self with "actual" preferences and makes the engagement decision. Once the user engages, System 1—the impulsive self—takes over, and the duration of engagement becomes independent of System 2’s preferences.
>
> This is exactly how we model inconsistent preferences. Kleinberg et al.’s framework, along with supporting references, directly informs our approach. Taken together, the empirical evidence for user strategizing and the two-mind model suggest that any strategic behavior arises at the engagement decision point (System 2), not in the engagement length (System 1).
>
> > How restrictive the special case in 3.4?
>
> We acknowledge that the special case presented in Sec. 3.4 is restrictive in that it limits content to two "types." However, this abstraction may be more general than it appears: in many settings, item types can be interpreted broadly, and with some approximation, users may consistently benefit more from one type than from another, more tempting alternative. We believe our closed-form results likely extend beyond this case, as the underlying structure often appears as a component of more complex patterns observed in practice. That said, the theoretical results may no longer be as elegant or interpretable outside this setting.
>
> > Is a "show less like this" button seen on a few platforms currently an example of a costly signal? What would motivate a platforms to implement a costly signaling feature?
>
> This is a great question—and in fact, the "show less like this" button was the original motivation for our research on how costly signaling can help! Regarding the burden on platforms, there is indeed a cost to them as well. As one can see in our special case, maximizing engagement alone does not incentivize the implementation of costly signals. While somewhat speculative, the shift from earlier social media platforms, which required more deliberate actions (e.g., clicks, menu options), to modern platforms featuring frictionless scrolling may reflect the platform’s disincentive to support such signaling mechanisms.
>
> **Modeling questions:**
>
> > On a session-level, why would the algorithm discount future rewards?
>
> We mainly view the possibility that the platform discounts future rewards as a theoretical curiosity that further generalizes our model. However, there are also practical scenarios where this assumption is reasonable. For instance, in time-sensitive campaigns, such as those with fixed deadlines, the timing of content consumption directly affects its value to the platform (for example, by going viral). Allowing for future-discounting algorithms also enabled us to uncover interesting insights—particularly regarding steerable sets, such as the observation in line 237: "as the algorithm becomes more foresighted, even slight disengagement from type 1 users can effectively influence its behavior."
>
> > When computing the algorithm's Q-value, why does the algorithm assume the users will continue to use the same strategy $f$ in the future?
>
> We assume that within a fixed session, users commit to a strategy $f$  that depends only on the content. While we acknowledge that users may adopt more complex strategies—such as those that vary over time within a session—we currently have no empirical evidence supporting such behavior. We therefore focus on a simpler strategy, which is (1) more plausible given typical user behavior, and (2) more feasible for the platform to learn. That said, this is certainly an assumption, and we recognize its limitations.
>
> > Is pulling the expected duration into the reward $r_\theta$ (in line 120) really without loss of generality?
>
> We have implicitly assumed that users discount rewards between interactions, but not during a single interaction. Under this assumption, line 120 holds without loss of generality. However, we acknowledge that if the duration of engagement is comparable to the time between interactions, it is indeed plausible that users also discount rewards within each engagement—introducing dependence to $\gamma_\mathcal{H}$. This would significantly complicate the analysis, as the distribution of engagement lengths (not just their expectation) would become relevant—an aspect our current model intentionally abstracts away for tractability—and likely only distorts the results negligibly. We thank the reviewer for noting this, and we will clarify the assumption more explicitly in the revised version.
>
> > What does "when users have consistent preferences ... the alignment problem reduces to engagement maximization" mean on page 1?
>
> By the mentioned sentence, we simply mean that if $r_\theta(s)$ and $1/\alpha_\theta(s)$ induce the same ordering over $s$, then the user and the algorithm have aligned interests. In this case, both aim to reach the same item $s^*_\theta = \arg\max_s r_\theta(s)$, which also maximizes the expected engagement length $1/\alpha_\theta(s)$. We had some difficulty understanding the example provided by the reviewer here. If the reviewer is referring to a scenario where a content $s$ yields a reward only after $T$ steps of interaction have passed, then while this is indeed interesting, it falls outside the scope of our model. In our setting, the platform’s reward—equivalently, the engagement length—depends only on the content and a time discount factor, not on when during the session (e.g., before or after time $T$) the content is consumed. On the other hand, if the reviewer means that the reward from a content $s$ is realized only if the user remains engaged for at least length of $T$, then the platform’s reward becomes $1/\alpha_\theta(s) = \mathbb{E}_\theta[1\\{y > T\\}]$, in which case our analysis still holds as described. We hope this addresses the reviewer’s question and would be happy to further clarify if needed.

---

> > ### Comment · Reviewer_7AFQ · 2025-08-05
> >
> > Thanks for your detailed response!
> >
> > I believe you may have misunderstood my question about what setting motivates the work-- I'm definitely on board with the premise that users strategically engage with content, but I was asking specifically about the model's assumption that the user chooses whether to engage with a particular item $s$ using System 2, but the duration of _engagement with $s$_ is determined by System 1. In the song-listening setting, this would be like deciding to listen to a song but System 1 deciding at what point in the song to skip to the next one, and I'm not sure whether this reflects a typical user's listening experience (I suppose the model would fit if you thought of $s$ as "the song and the stream of songs the platform autoplays after $s$", perhaps this is what you mean). By contrast, Kleinberg et al. assume System 1 chooses how many time steps to remain on the platform, but during this time the user will be receiving different pieces of content. If I've misunderstood the model, please correct me! In general, I don't have an issue with the recommendation model not-quite-fitting all real-world recommendation settings, because platforms all differ from one another. However, ideally there is some platform or set of platforms that do function very similarly to the model.
> >
> > Apologies that my example for the last modeling question was confusing! I meant the second thing, that there might be a piece of content A where a rational user gets a utility of $r_A$ after engaging with the content for $T$ time steps, and another piece of content B where the user gets utility of $r_B > r_A$ after only one time step. The platform would only ever show content A to the user because the platform gets utility $T > 1$, but the user would only get utility $r_A < r_B$. I believe the confusion comes from the phrase "consistent preferences", which I was taking to mean that "System 1 is no longer controlling engagement", but from your explanation it seems that your intention was to convey that "The engagement length is in proportion to the reward (or at least it induces the same ordering)". I think would be helpful to clarify this terminology in the paper.

---

> > > ### Author Response · Authors · 2025-08-05
> > >
> > > Thanks for further clarifying your point. We believe we now understand it more clearly. Since we’ve already discussed the music listening example, let’s work that out. First, all of your interpretations are actually correct: we can think of $s$ as the type of playlist or the entry song, and the length of engagement as the number of songs played from the playlist. We can even go further and interpret $s$ as the specific song, with the engagement length depending on how many interruptive ads system 1 tolerates (we can also see these decisions in our chatbot example, where the "modes" represent types: deciding whether to engage with a psychologist chatbot in an affirmative mode is a system 2's decision, but the duration of the conversation is determined by system 1).
> > >
> > > Just to clarify the connection to Kleinberg et al.: we treat the entire interaction that system 1 has after system 2 decides to engage as a single item. While the user may consume different things during engagement, from system 2’s perspective, the only relevant parameter is the expected reward, and from the platform’s perspective, it’s the expected length of engagement. What exactly gets consumed during the engagement does not reveal information about the user’s type, since it wasn’t the result of a rational/strategic decision and the platform only records engagement decisions for simplicity. As a result, we can abstract away that complexity and treat the interaction as a single item (such as the first song or playlist that system 2 chooses to engage with). Note that, as you have thoughtfully asked in the questions, we are assuming that the user does not discount rewards during engagement in this abstraction.
> > >
> > >
> > > > from your explanation it seems that your intention was to convey that "The engagement length is in proportion to the reward (or at least it induces the same ordering)". I think would be helpful to clarify this terminology in the paper.
> > >
> > > And thanks for clarifying your last question. Yes, this is exactly what we meant, and we will clarify this terminology in our revision of the introduction, where the confusion came from. Thank you for your accurate read of the paper again.

---

> > > > ### Comment · Reviewer_7AFQ · 2025-08-07
> > > >
> > > > Thanks for these clarifications!

---

### Note · Authors · 2025-08-12

We thank the reviewers for their insightful engagement with our paper. We are glad that reviewers found our **theoretical contributions to be rigorous and non-trivial**, noting that our results are interesting (7AFQ, 8uFD) and that our framework offers “a rigorous theoretical framework” with “well-defined guarantees” (HcaA). Reviewers also emphasized the **novelty, positioning, and potential applications of our work**, describing it as “a novel bridge” between strategic and inconsistent content consumption literatures (7AFQ), recognizing its connections to both theory and practice (bpFx), appreciating the range of extensions such as costly signaling (bpFx), and highlighting its potential to shed light on real-world phenomena (HcaA). We’re also pleased that the reviewers recognized the **clarity and quality** of our paper with three reviewers finding the quality of our paper excellent.

We used the rebuttal to further clarify the settings that motivate our theoretical framework (7AFQ), our modeling assumptions (7AFQ, bpFx), and additional connections to practice and policy. For example, we discussed how giving users just one additional option—even one that simply burns effort—can significantly improve their ability to steer the algorithm (bpFx), and when and how middleware tools (as discussed by Fukuyama et al.) might be used to extend users’ effective horizons when their own is insufficient for alignment (8uFD). We were also pleased that, through their deep engagement with the paper, reviewers connected our theory to practical scenarios, such as how a small “show less like this” feature can serve as costly signaling (7AFQ), how the framework applies to doomscrolling (HcaA), and how it could be further extended to study other phenomena.

We thank the reviewers once again for their thoughtful feedback and constructive suggestions. We believe their engagement has helped us better articulate the motivations, assumptions, and broader relevance of our framework, and appreciate their recognition of our contributions.

---

### Decision · Program_Chairs · 2025-09-17

**Decision:**

Accept (poster)

**Comment:**

The authors examine a hypothetical recommender system where users display multiple behavior profiles, and interact with the recommender system over a sequence of item selections. They characterize the possibility of the user steering the algorithm towards its preferences in terms of the length of the interaction horizon, or burden of alignment, as the authors put it. Reviewers were excited about the motivation of the paper, the non-trivial theoretical results it provides, and potential for follow-up work. However some concerns about the strength of the empirical results were raised, along with uncertainties about the relevance to practical recommender systems. Overall, I recommend accepting the paper, and encourage the authors to integrate the feedback from the reviewers, with a special focus on properly contextualizing the results so that future work—especially those focused on empirics—can effectively extend the key ideas of the paper.